# Alleviation of Memory Deficit by Bergenin via the Regulation of Reelin and Nrf-2/NF-κB Pathway in Transgenic Mouse Model

**DOI:** 10.3390/ijms22126603

**Published:** 2021-06-20

**Authors:** Bushra Shal, Adnan Khan, Ashraf Ullah Khan, Rahim Ullah, Gowhar Ali, Salman Ul Islam, Ihsan ul Haq, Hussain Ali, Eun-Kyoung Seo, Salman Khan

**Affiliations:** 1Pharmacological Sciences Research Lab, Department of Pharmacy, Faculty of Biological Sciences, Quaid-i-Azam University, Islamabad 45320, Pakistan; bushra.shal@gmail.com (B.S.); adkhan165sbbu@gmail.com (A.K.); ashrafwazir6@gmail.com (A.U.K.); 2Department of Pharmacy, University of Peshawar, Peshawar 25120, Pakistan; Rphrahimullah@gmail.com (R.U.); gowhar_ali@uop.edu.pk (G.A.); 3School of Life Sciences, College of Natural Sciences, Kyungpook National University, Daegu 41566, Korea; dr_ssulman@yahoo.com; 4Department of Pharmacy, Faculty of Biological Sciences, Quaid-i-Azam University, Islamabad 45320, Pakistan; ihsn99@yahoo.com (I.u.H.); h.ali@qau.edu.pk (H.A.); 5College of Pharmacy, Graduate School of Pharmaceutical Sciences, Ewha Womans University, Seoul 03760, Korea

**Keywords:** memory loss, Alzheimer’s disease, Bergenin, Reelin signaling, oxidative stress, neurodegeneration

## Abstract

The present study aims to determine the neuroprotective effect of Bergenin against spatial memory deficit associated with neurodegeneration. Preliminarily, the protective effect of Bergenin was observed against H_2_O_2_-induced oxidative stress in HT-22 and PC-12 cells. Further studies were performed in 5xFAD Tg mouse model by administering Bergenin (1, 30 and 60 mg/kg; orally), whereas Bergenin (60 mg/kg) significantly attenuated the memory deficit observed in the Y-maze and Morris water maze (MWM) test. Fourier transform-infrared (FT-IR) spectroscopy displayed restoration of lipids, proteins and their derivatives compared to the 5xFAD Tg mice group. The differential scanning calorimeter (DSC) suggested an absence of amyloid beta (Aβ) aggregation in Bergenin-treated mice. The immunohistochemistry (IHC) analysis suggested the neuroprotective effect of Bergenin by increasing Reelin signaling (Reelin/Dab-1) and attenuated Aβ (1–42) aggregation in hippocampal regions of mouse brains. Furthermore, IHC and western blot results suggested antioxidant (Keap-1/Nrf-2/HO-1), anti-inflammatory (TLR-4/NF-kB) and anti-apoptotic (Bcl-2/Bax/Caspase-3) effect of Bergenin. Moreover, a decrease in Annexin V/PI-stained hippocampal cells suggested its effect against neurodegeneration. The histopathological changes were reversed significantly by Bergenin. In addition, a remarkable increase in antioxidant level with suppression of pro-inflammatory cytokines, oxidative stress and nitric oxide production were observed in specific regions of the mouse brains.

## 1. Introduction

Dementia is a general term that illustrates the cognitive decline in brain function. According to the recent literature, Alzheimer’s disease (AD) is the most common type of dementia reported in more than 35 million people globally, leading to death within 3 to 9 years after diagnosis [1]. With a rapid increase in aged populations, dementia and cognitive decline are affecting the healthcare systems worldwide [1]. The toxic events involved in the pathogenesis of AD are characterized by the aggregation of amyloid plaques (Aβ) in the brain, composed mainly of 40–42 amino acid amyloid peptides resulting in synaptic and neuronal loss [2]. Due to a lack of disease-modifying therapy, there is a need for introducing a novel therapeutic approach. Over the years, therapeutic strategies for the removal of plaque and decreasing extracellular amyloid deposition have dominated research efforts in AD. 

Evidence suggests that proteolysis of transmembrane amyloid precursor protein (APP) generates Aβ peptides, most commonly 42-residue-long form (Aβ-42) that plays a central role in the pathogenesis of AD [3]. Accumulation of extracellular amyloid plaques leads to neurodegeneration in the hippocampal and cortex region, resulting in activation of inflammatory and oxidative stress pathways accompanied by memory deficits and cognitive impairment [4]. These Aβ peptides aggregate to form oligomeric species resulting in synaptic loss either by directly blocking or/ abnormally stimulating receptors or by affecting the levels of proteins that maintain or increase synaptic plasticity [5]. The signaling pathways that might contribute towards Aβ-induced synaptic plasticity impairment include the Reelin-signaling pathway [5]. Reelin overcomes the toxicity of Aβ by interacting with Aβ-42-soluble species sequestering in amyloid plaques, controls APP processing, decreases Tau phosphorylation and enhances cognitive performance [6]. Reduction of Reelin in AD accelerates the onset of plaque formation, accompanied by activation of glial cells, particularly microglia [7]. 

Microglial activation by aggregated Aβ may be favorable for the clearance of Aβ peptides [8]. Neurotoxicity in relation to microglial activation depends on increased functional activity of innate immune receptors such as Toll-like receptors-4 (TLR-4), resulting in the activation of nuclear factor-kappa light chain enhancer of B-cells (NF-κB) producing neurotoxic mediators, causing neuronal cell death [9]. An imbalance in the production of reactive oxygen species (ROS) and its removal leads to oxidative stress suggesting a compromised defense mechanism balanced by nuclear factor erythroid-2 related factor (Nrf-2) and heme-oxygenase-1 (HO-1), thus decreasing antioxidant enzymes [10]. 

Recent studies have suggested the involvement of nitric oxide (NO) either directly or indirectly in neuronal death in AD [11]. Increased neurotoxic effects of NO might be mediated by the activation of intracellular signaling cascades via oxidative damage. In particular, the reaction of NO with superoxide anion (O2-) at sites of plaques generates a strong oxidant, i.e., peroxynitrite (ONOO-), which is capable of inducing neuronal cell damage [12]. Elevated oxidative and nitrosative stress may cause cellular abnormalities; the free radicals could destroy cell membrane potential through lipid peroxidation, increasing structural protein misfolding and aggregation [13,14]. In response to the Aβ aggregation and elevation of ROS in the apoptotic pathway, the neurons become activated. Apoptosis plays a key role in neuronal cell death through activation of the apoptotic signaling pathway (Bax/Bcl-2/caspase-3). Upon receiving a death signal, Bax translocates into the mitochondria and interacts with anti-apoptotic Bcl-2 family members to control the progression of apoptosis and activates caspase-3 [15]. 

In order to explore disease-modifying therapies and drugs that target specific molecular pathways and block the progression of dementia and cognitive decline in AD, there is a need for screening natural compounds [16]. Several natural compounds have shown therapeutic potential, characterized by multiple pharmacological properties effective against spatial memory deficit [17]. 

In the present study, various in vitro and in vivo techniques were employed to get a better understanding of Bergenin against disease progression and prevention. Bergenin is an isocoumarin derived from the rhizome of *Bergenia ciliata*; it belongs to the family Saxifragaceae. It has been reported to have antioxidant, anti-inflammatory, antiarthritic, immunomodulatory, antinarcotic, wound healing, and antidiabetic properties [18]. The present study aimed to determine the effect of Bergenin against spatial memory deficit in the 5xFAD Tg mouse model. Its neuroprotective effect was initially evaluated in HT-22 and PC-12 cells. Promising results obtained from the in vitro screening resulted in further investigation into the transgenic mouse model. The present study suggested the neuroprotective effect of Bergenin against Aβ aggregation in 5xFAD Tg mice by regulating Reelin signaling pathway and decreasing oxidative stress, neuro-inflammation, and apoptosis to alleviate the spatial memory deficit associated with AD. 

## 2. Results

### 2.1. Bergenin Exerts Neuro-Protective Effect in HT-22 and PC-12 Cells

An MTT assay was performed to investigate the cytoprotective effect of Bergenin against H_2_O_2_-induced oxidative stress in HT-22 and PC-12 cells. Cells exposed to 200 µM of H_2_O_2_ resulted in a significant (*p* < 0.001) reduction in cell viability compared to the cells with no H_2_O_2_ exposure, i.e., control cells. The pretreatment with Bergenin (50 and 100 µM) remarkably (*p* < 0.01, *p* < 0.001) increased cell viability in HT-22 and PC-12 cells compared to the cells treated only with H_2_O_2_ as shown in Figure 1B,G. 

### 2.2. Bergenin Reduces Nitrite Production in HT-22 and PC-12 Cells

Nitrite production was observed to be significantly (*p* < 0.001) increased in cells incubated with H_2_O_2_ (200 µM) alone compared to control cells. Dose-dependent attenuation of NO production was observed with Bergenin pretreatment in HT-22 and PC-12 cells shown in Figure 1C,H. Pretreatment with Bergenin (50 and 100 µM) significantly (*p* < 0.01, *p* < 0.001) decreased nitrite production in HT-22 and PC-12 cells compared to the cells treated with H_2_O_2_ alone. Curcumin (10 µM) was used as a positive control and also significantly decreased the NO produced compared to the H_2_O_2_-alone cells.

### 2.3. Bergenin Enhances Anti-Oxidants and Decreases Oxidative Stress in Cells

The antioxidant enzymes and proteins, such as reduced glutathione (GSH) and superoxide dismutase (SOD), decreased significantly (*p* < 0.001) in H_2_O_2_ alone-treated HT-22 cells compared to the control cells (only DMSO). However, pretreatment with the Bergenin dose-dependently enhanced antioxidant level as compared to the cells treated with H_2_O_2_ alone. The GSH level was increased significantly in the Bergenin-treated cells (50 µM (*p* < 0.01) and 100 µM (*p* < 0.001) compared to H_2_O_2_ alone (Figure 1D). Similarly, the SOD level was also raised significantly in the Bergenin-treated cells 10 µM (*p* < 0.05), 50 µM (*p* < 0.01) and 100 µM (*p* < 0.001) as compared to H_2_O_2_ alone (Figure 1E). However, a significant decrease in oxidative stress was observed by decreased level of MPO in HT-22 cells at a dose of 50 µM and 100 µM (*p* < 0.001) (Figure 1F). 

Similarly, the level of glutathione was observed to significantly decreased in the H_2_O_2_ alone-treated PC-12 cells compared to the control cells (only DMSO). The pretreatment with Bergenin remarkably enhanced the antioxidant level at 50 µM (*p* < 0.01) and 100 µM (*p* < 0.001) compared to H_2_O_2_ alone (Figure 1I).

### 2.4. Bergenin Enhances Spatial Memory in 5xFAD Tg Mice

Spatial memory deficit in 5xFAD Tg mice was evidenced by its prolonged stay in Y-maze, displaying a remarkable (*p* < 0.001) increase in escape latency compared to the wild-type-saline-treated group. Treatment with Bergenin and Galanthamine for five consecutive days improved spatial memory. A dose-dependent improvement in escape latency and spontaneous alteration behavior (SAB) was observed with Bergenin. Bergenin (60 mg/kg) remarkably (*p* < 0.001) reduced the escape latency throughout as compared to the 5xFAD Tg-saline-treated animals shown in Figure 2A. The 5xFAD Tg-saline-treated mice strain displayed a significant decrease in the percent spontaneous alteration as compared to the wild-type-saline-treated mice. However, this reduction in % SAB was remarkably (*p* < 0.001) reversed by Bergenin (60 mg/kg), as shown in Figure 2B.

The results obtained from the Morris water maze test (MWM) also displayed an increased escape latency (*p* < 0.01, *p* < 0.001) in 5xFAD Tg mice compared to the wild-type-saline-treated group. Treatment with Bergenin (30 mg/kg) decreased escape latency on 5th day (*p* < 0.01), while at a dose of 60 mg/kg Bergenin remarkably (*p* < 0.05, *p* < 0.01 and *p* < 0.001) reduced latency time on days 3 to 5. Additionally, the probe trals showed that 5xFAD Tg mice spent less time (*p* < 0.001) in the target quadrant compared to the wild-type-saline-treated mice. However, a significant alleviation of spatial memory deficit was observed in the Bergenin-treated group (60 mg/kg), displaying more time spent (*p* < 0.01, *p* < 0.001) in the target quadrant compared to the 5xFAD Tg mice shown in Figure 2.

### 2.5. Effect of Bergenin on Locomotor and Anxiety-Like Behavior in 5xFAD Tg Mice

Open field test (OFT) was performed for the assessment of locomotor activity, anxiety, and exploratory behavior in mice. In the present study, OFT parameters such as time spent in the center, rearing behavior, and a number of line crossings were observed (Figure 3A–C). The 5xFAD Tg mice group exhibited a reduced number of rearing, number of line crossings, and time spent in the center as compared to the wild-type-saline-treated group. Whereas, Bergenin dose-dependently improved all the behavioral parameters, including the number of rearings, number of line crossings, and time spent in the center of OFT compared to the 5xFAD Tg group. 

### 2.6. Bergenin Protects against Decrease in Proteins, Lipids and Their Secondary Derivatives

The band assignments and intensities reflect the biomolecular composition of hippocampal tissues in the IR spectra (Figure 3D). The effect of Bergenin treatment on 5xFAD Tg mice was investigated using FT-IR spectroscopy using different functional groups (Figure 3E–I). The FT-IR absorption spectra were obtained between the 4000 and 500 cm^−1^ regions (Figure 3E). The changes in the peak intensities of the macromolecules such as lipids, proteins, phospholipids, and nucleic acid (Figure 3F–I) are represented in Table 1. Detailed spectral analyses were performed in different frequency ranges, i.e., ~3600–3100 cm^−1^ (Figure 3F), and ~1800–1500 cm^−1^ (Figure 3G), to understand the details of proteins and lipids. In addition to this, secondary derivatives of the lipid region (~3050–2800 cm^−1^) (Figure 3H) and amide-I region (~1700–1600 cm^−1^) (Figure 3G) were analyzed for the secondary structures of proteins and lipids in mouse brain tissues. The changes in phospholipids and nucleic acid content (~1300–1000 cm^−1^) (Figure 3I) were also determined.

The band intensities in the FT-IR spectra of amide-A (~3600–3100 cm^−1^), Amide-I, and Amide-II (~1800–1500 cm^−1^) proteins in mouse brain tissues were observed to be significantly decreased in the Tg mice, indicating a decrease in protein content compared to the wild-type control, while Bergenin treatment reversed the band area towards the control group indicated in Table 1. Furthermore, to investigate the changes in lipid composition of brain tissue, the spectral region from ~3050–2800 cm^−1^ was considered (Figure 3H). The bands obtained in this region were due to the absorption of olefinic (=C-H), CH3, and CH2 stretching groups (Figure 3H). The 5xFAD Tg group showed a significant decrease in the band intensity throughout the groups compared to the control group. However, treatment with Bergenin showed appreciable restoration of lipids and proteins in the brain tissue compared to the negative control group (Table 1). Similarly, Bergenin also significantly restored the phospholipids, and nucleic acid content decreased in the transgenic group (Figure 3I).

The lipid/protein ratio of the secondary derivative within the range of ~1800–1500 cm^−1^, indicated protein aggregation at the spectral range of ~1630–1625 cm^−1^ representing an increase in β-sheet in 5xFAD Tg mice (Figure 3G). The spectral analysis of the Bergenin-treated group displayed a reduction in absorbance intensity compared to the 5xFAD Tg group (Table 2). 

### 2.7. Bergenin Improves Histopathology in H&E Stained 5xFAD Tg Mice Brain Tissues

The neuronal cells in the H&E-stained sections of hippocampal dentate gyrus (Figure 4A) and prefrontal cortex (Figure 4B) of saline-treated wild-type group (Figure 4(Aa),(Ba)), and Bergenin-treated groups (Figure 4(Ac),(Bc)) were well arranged. However, the brain sections of 5xFAD Tg-saline-treated mice (Figure 4(Ab),(Bb)) displayed neuronal damage in the granular layer of dentate gyrus of the hippocampal region and granular layer of the prefrontal cortex Bergenin showed marked protection against neuronal damage in 5xFAD Tg mice, as shown in Figure 4C,D.

### 2.8. Bergenin Attenuates Amyloid Plaque Deposits in 5xFAD Tg Mice Brain Tissues

Amyloid plaque deposits were observed in 5xFAD Tg mice compared to the control animals in all the regions of interest (ROI), i.e., HC, EC, and PFC (Figure 5A–C). Treatment with Bergenin (60 mg/kg) displayed a reduced amyloid burden in all the ROI of mice brain in comparison to the 5xFAD Tg group (Figure 5). Amyloid loads quantified in different regions of interest showed a significant (*p* < 0.001) reduction in the Bergenin-treated groups as compared to the 5xFAD Tg-saline-treated animals as shown in Figure 5D–F.

### 2.9. Bergenin Decreases Clusters of PAS-Positive Granules in 5xFAD Tg Mice Brain Tissues

Extracellular PAS granules were analyzed in PAS-stained multiple ROI in mice brains, i.e., HC (Figure 6A), EC (Figure 6B), and PFC (Figure 6C). No granules were seen in the brain sections of control animals whereas, a number of different-sized granules were observed in 5xFAD Tg mice in all the regions of interest (Figure 6(Ab),(Bb),(Cb)). However, treatment with Bergenin (60 mg/kg) displayed a marked reduction in the PAS-positive granules in all ROI under study, i.e., (Figure 6(Ac),(Bc),(Cc)). The PAS-positive granules in all the regions were indicated by arrows. Quantitative analysis showed a remarkable reduction in granular load in Bergenin-treated (60 mg/kg) groups as compared to the 5xFAD Tg-saline-treated group, as shown in Figure 6D–F. 

### 2.10. Bergenin Increases Reelin Signaling Pathway and Attenuates Aβ (1-42) 

Immunohistochemical (IHC) analysis of transgenic mice showed a decreased Reelin expression in the hippocampal region of mouse brains compared to the wild-type control (Figure 7(Ab)). Dab-1 adaptor protein expression was also found suppressed in 5xFAD Tg mice (Figure 7(Bb)), along with an enhanced Aβ (1-42) aggregation in the transgenic control group (Figure 7(Cb)). Treatment with Bergenin (60 mg/kg) significantly (*p* < 0.001) enhanced the expression of Reelin and Dab-1, while attenuates Aβ (1-42) aggregation compared to the transgenic control group (Figure 7D–F).

### 2.11. Effect of Bergenin on DSC Analysis

DSC measurement showed large differences between the heat flow profiles for wild-type and 5xFAD Tg mice in the hippocampal regions of mouse brains (Figure 7G). The heat flow profiles of the hippocampal region displayed endothermic denaturational transitions presented in 5xFAD Tg mice. The treatment with Bergenin displayed a well-preserved heat flow typical to the wild-type mice.

### 2.12. Effect of Bergenin on Nrf-2, NF-κB and Bax Signaling Proteins 

To determine the effect of Bergenin against spatial memory deficit in AD, western blot analysis was performed in the hippocampal regions of mouse brains (Figure 8A), uncropped images were shown in Appendix A. The results suggested a significant (*p* < 0.001) increase in protein expression of Nrf-2 (Figure 8B) as compared to 5xFAD Tg group. In addition, a remarkable (*p* < 0.001) decrease in the degradation of IκB-α (Figure 8C) and prevented NF-κB translocation to the nucleus (Figure 8D) was observed. The treatment with Bergenin also significantly (*p* < 0.001) upregulated Bcl-2 pro-apoptotic protein (Figure 8E) compared to 5xFAD Tg mice group. 

### 2.13. Bergenin Decreases Inflammatory and Oxidative Stress Pathway

Immunohistochemical (IHC) analysis demonstrated the neuroprotective effect of Bergenin against inflammatory and oxidative stress pathways. Treatment with Bergenin (60 mg/kg) demonstrated a remarkable suppression in the relative expression of TLR-4 (Figure 9A) (*p* < 0.001) and NF-κB (Figure 9C) (*p* < 0.001) along with an increase in the relative expression of IκB-α (Figure 9B) in the hippocampal region compared to the transgenic-saline control group. Moreover, Bergenin (60 mg/kg) also presented a significant decrease in the protein expression of Keap-1 (Figure 9D) (*p* < 0.001) and an increased expression of Nrf-2 (Figure 9E) (*p* < 0.001) and HO-1 (Figure 9F) (*p* < 0.001) proteins in the hippocampal region compared to the transgenic-saline control group shown in Figure 9.

### 2.14. Bergenin Reduces Apoptotic Pathway

Immunohistochemical analysis demonstrated a protective effect of Bergenin against apoptotic pathways. IHC staining for apoptotic markers presented a significant downregulation of Bax (Figure 10B) and cleaved casp-3 (Figure 10C) (*p* < 0.001) expression, with an enhanced relative expression of Bcl-2 (Figure 10A) (*p* < 0.001) in hippocampal regions of 5xFAD Tg-Bergenin-treated group compared to control group as shown in Figure 10. Furthermore, the level of apoptosis in a larger population of hippocampal neurons was determined by flow cytometer analysis. The neuronal cells stained with Annexin V/FITC and PI were sorted into apoptotic and non-apoptotic populations (Figure 10B). The hippocampal neuronal cells from 5xFAD Tg mice showed increased apoptosis with a total apoptosis rate of 74.12%, displaying an increased population of apoptotic cells and decreased live cell population compared to the wild-type group (3.26%). However, treatment with Bergenin showed a reduced apoptosis rate to 5.20% presenting a significant reduction in apoptotic cell population compared to the control group. Additionally, quantification for (%) early apoptosis, (%) late apoptosis and (%) live cells were done for wild-type, 5xFAD Tg and Bergenin-treated group shown in Figure 10(Da–c).

### 2.15. Bergenin Enhances Antioxidant Levels in 5xFAD Tg Mice Brain 

The antioxidant status, such as GSH, GST, SOD, and Catalase level in HC, PFC and EC regions of transgenic-saline-treated mice were significantly (*p* < 0.001) decreased compared to the wild-type-saline-treated control. Bergenin (60 mg/kg) significantly reversed the reduced activity of GSH (Figure 11A–C), GST (Figure 11D–F), SOD (Figure 11G–I), and Catalase (Figure 11J–L) in all the regions of interest in mouse brains, resulting in free radical scavenging compared to 5xFAD Tg-saline-treated group. 

### 2.16. Bergenin Reduces Oxidative Stress and Nitrite Production in 5xFAD Tg Mice Brain

The level of oxidative stress markers in different regions of interest was measured in mouse brains. The lipid peroxidase (LPO) and myeloperoxidase (MPO) levels in HC, PFC and EC regions of 5xFAD Tg in mouse brain were significantly (*p* < 0.001) higher than the wild-type-saline-treated mice. At the same time, a reduction in tissue MDA level (Figure 12A–C) and MPO level (Figure 12D–F) was observed with Bergenin in all regions of interest compared to 5xFAD Tg mice. Nitric oxide (NO), an important mediator of inflammation, was observed to be significantly (*p* < 0.001) higher in 5xFAD Tg mice compared to the wild-type-saline control. The animals treated with Bergenin (60 mg/kg) showed a significant (*p* < 0.001) reduction in NO production (Figure 12G–J) both in tissue in plasma as compared to the 5xFAD Tg-saline-treated mice. 

### 2.17. Bergenin Reduces Pro-Inflammatory Cytokine Production in Mice Brain

The production of pro-inflammatory cytokines was assessed in HC, EC and PFC regions of mouse brains (Figure 12K–P). A remarkable (*p* < 0.001) increase in IL-1β (Figure 12K–M) and TNF-α (Figure 12N–P) in 5xFAD Tg mouse brain regions were observed compared to wild-type healthy mice. The treatment with Bergenin (60 mg/kg) significantly (*p* < 0.001) decreased the level of pro-inflammatory in all the regions of interest as compared to 5xFAD Tg mice.

### 2.18. Molecular Docking Analysis with Bergenin

Bergenin showed remarkable interaction with multiple protein targets such as Keap-1, Nrf-2, HO-1, TLR-4, IκB-α, NF-κB, Bcl-2, Bax, Caspase-3 and Amyloid beta (Aβ). The two-dimensional and three-dimensional structural arrangements of the best-docked poses were shown in Figure 13A. The binding energies of various proteins after docking with Bergenin were calculated as −7.8 Kcal/mol for 1U6D, −8.3 Kcal/mol for 2FLU, −6.6 Kcal/mol for 1irm, −7.2 Kcal/mol for 3vq2, −7.2 Kcal/mol for 6y1j, −8.8 Kcal/mol for 1vkx, −7.7 Kcal/mol for 2w3l, and −5.8 Kcal/mol for 5w62 shown in the heatmap (Figure 13B). Table 3 showed binding energies and binding residues of protein targets docked with Bergenin. Furthermore, Bergenin formed six hydrogen bonds with Keap-1 and Caspase-3, four hydrogen bonds with IκB-α, NF-κB, and Bax, three hydrogen bonds with Nrf-2, HO-1, and TLR-4. Two hydrogen bonds were observed with Bcl-2 and Amyloid beta (Aβ). In addition, hydrophobic interactions were also observed, which further provided strength to the protein/ligand binding complex. Bond distance and corresponding polar and electrostatic interactions were presented in a two-dimensional (2D) ligplot arrangement shown in Figure 14. Analysis of these binding interactions of Bergenin revealed that the compound was efficiently docked inside the active site of the target protein structures.

## 3. Discussion

Memory impairment and cognitive decline are associated with AD, a well-known neurological disorder. The toxic events involved in the pathogenesis of spatial memory deficit in AD include multiple biological processes such as Aβ aggregation, neurofibrillary tangles formation, neuroinflammation, and oxidative stress linked to neurodegenerative diseases [19]. Inhibition of Aβ aggregation is a promising therapeutic approach towards the identification of disease-modifying therapy against AD. The current study aimed to determine the neuroprotective activity of Bergenin that may result in the identification of a novel therapeutic compound with an ability to combat spatial memory disorder. The initial screening of the natural compound revealed its neuroprotective ability against H_2_O_2_-induced oxidative stress in HT-22 and PC-12 cells. Evidence has indicated etiological links between H_2_O_2_ generation and neurodegenerative disorder, neural cells exposed to H_2_O_2_ undergo apoptosis-like delayed death. The treatment of oxidative stress by increasing cellular processes that decrease production or removes ROS has been considered a therapeutic target for the treatment of neurological diseases [20]. 

Promising results obtained from the preliminary study resulted in further investigation of its neuroprotective ability in 5xFAD Tg mouse model. The genetically engineered mouse models have contributed significantly to the elucidation of the underlying mechanism of cognitive decline and memory impairment associated with AD [21]. Although none of the models completely represents the phenotypes present in AD patients, however, similar changes can be observed related to the disease progression and cognitive decline [22]. The 5xFAD transgenic mice models are widely used to determine the potential therapeutic candidate. This mouse model has five different FAD mutations in both amyloid precursor protein (APP) and presenilin, displaying higher levels of Aβ accelerating plaque formation and memory impairment [23]. The present study aimed to determine detailed neuropathology, behavior and histological analysis of 5xFAD Tg mouse model. 

Memory-related behaviors were determined by using Y-maze and MWM paradigm. In the Y-Maze test, Bergenin dose-dependently improved the spatial memory deficit determined by a decrease in escape latency and increase in spontaneous alteration behavior, respectively, as compared to the 5xFAD Tg mice group. The MWM test has been widely used for the assessment of spatial memory, based on the assumption that animals have acquired an optimal approach to explore their environment and escape from the water with the least effort [24]. The findings of the present study suggested a cognitive decline in the 5xFAD Tg mice group, that required more time to find the escape platform in training trials and spent less time in the target quadrant during the probe trials as compared to the wild-type group. However, treatment with Bergenin significantly improved the spatial memory, displayed a decrease in the latency time during training trials, and increased time spent in the target quadrant during the probe trials. Additionally, OFT was performed to determine the locomotor (number of line crossings), anxiety (time spent in the center) and exploratory behavior (number of rearings) in the 5xFAD Tg mice [25]. Bergenin dose-dependently improved all the behavior parameters compared to the 5xFAD Tg mice; these results were consistent with previous findings [26,27]. 

The extracellular aggregation of Aβ in neuritic plaques and its binding to a variety of receptors appears to be the characteristic pathological hallmark of AD. In addition, Aβ oligomers were proposed to induce mitochondrial dysfunction and oxidative stress resulting in neuronal toxicity [28]. The brain is one of the most lipid-rich organs and the correct lipid turnover or metabolism is important for its proper functioning. Evidence suggests that lipids play a vital role in composition of the brain and influences its neuronal functioning, including behavior [29]. Alterations in the lipid, protein, or phospholipids could result in the pathological processes in the brain. The aggregation of Aβ results in the generation of free radicals such as ROS that reacts rapidly with proteins and lipids results in the production of toxic oxidized proteins and lipid peroxides [30]. In the present study, FT-IR spectroscopy provided structural information of biomolecules such as proteins, lipids and phospholipids, that allowed detection of changes in the macromolecular structures in the hippocampal regions of 5xFAD Tg mouse brains [31,32,33]. The FT-IR spectra demonstrated changes in the lipid content of the brain tissue displaying changes in lipid acyl (CH2), methyl group (CH3), olefinic (=CH) and lipid ester (C=O). The increased oxidative stress leads to lipid degradation producing smaller fragments [34]. The decrease in unsaturated lipids, olefinic band and saturated lipids CH2 symmetric bands in our study revealed the free radical damage of the lipid molecules in transgenic mouse brains. However, an increase in lipid esters (C=O) was found, emphasizing the involvement of lipid peroxidation in transgenic mouse brains. Phospholipids are one of the major components of brain membranes, hydrolyzed into secondary messengers that mediates acute responses. Treatment with Bergenin significantly restored phospholipids and nucleic acid content compared to the 5xFAD Tg group.

FT-IR spectra have been shown to be sensitive towards the identification of secondary structures of proteins, making it valuable towards the identification of protein aggregation. The amide-A region mainly composed of N-H protein stretching displayed a decrease in the intensity of the band, indicating a decrease in the protein concentration in 5xFAD Tg mouse brains in comparison to the wild-type control. The treatment with Bergenin remarkably restored the band intensity compared to the Tg mice group. The two prominent regions of protein FT-IR spectra are Amide-I and Amide-II bands mainly composed of C=O stretching, N-H bending, and C-N stretching were found to be decreased in 5xFAD Tg group. Treatment with Bergenin prevented the decrease in the concentration of protein. Oxidative stress caused loss of protein due to breakdown of sulphide and hydrogen bonds in proteins, resulting in the unfolding of proteins and disorientation of their internal structures and accumulation in specific tissue regions [35]. The Amide-I band mainly consisted of secondary structures such as parallel and anti-parallel β-sheets [36]. In an anti-parallel β-sheet, the amide-I region displayed a major component producing a high-intensity band at ~1630 cm^−1^ while, a minor component displayed a low-intensity band at ~1695 cm^−1^. The amide-I region for parallel β-sheet displayed only a major component at ~1630 cm^−1^ [37,38]. The lipid/protein ratio displayed an increase in β-sheet content in 5xFAD Tg group, which was significantly restored by Bergenin treatment preventing the aggregation process and suggesting a protective effect against neurodegeneration [39].

Reelin, a glycoprotein of extracellular matrix is crucial for neural development that regulates the development and plasticity of cerebral cortex, influencing the synaptic neurotransmissions and memory in the adult brain [40]. In relation to the AD pathology, Reelin present in amyloid plaques controls APP processing and counteracts Aβ-induced synaptic impairment [41]. The reduction of Reelin in AD mouse model displays an increased onset of plaque formation [42]. Reelin binds to its receptor and relays signals into the cell via Dab-1 adapter (Disabled-1), preventing Tau hyperphosphorylation [43]. The present study demonstrated the prevention of Aβ (1-42) aggregates in the hippocampal region by restoring Reelin signaling pathway in the Bergenin-treated group compared to 5xFAD Tg mice. 

In the present study, DSC analyses were performed to characterize the thermodynamic properties of amyloid fibrils highlighting the thermal fluctuations of proteins during their conversion into amyloid fibrils. Higher temperatures may result in the production of secondary structures and compaction of intrinsically disordered proteins (IDP), leading to the production of endothermic peak [44]. According to the reported data, the IDP aggregates in AD are mostly amyloid plaques producing neurotoxic effects by diffusing brain parenchyma and reaching synaptic clefts resulting in synaptic malfunctions that cause neuronal and cognitive dysfunction [45]. The endothermic events observed with DSC scans in this study were indicative of protein aggregation. The aggregation of proteins typically occurs above their denaturation temperature resulting in protein unfolding leading to random coil conformations [44]. It has been suggested that the endothermic heat flow can be accompanied by the formation of amyloid fibrils. The exothermic transition observed in the current study represented healthy animals, which was missing in 5xFAD Tg group, while treatment with Bergenin preserved the exothermic transition. 

Numerous studies have proposed the involvement of oxidative and nitrosative stress in the pathogenesis of AD, in relation to the presence of amyloid plaques [4]. The Nrf-2/Keap-1 signaling pathway has an important role in the cellular defense and maintenance of physiological balance against oxidative stress [46]. Interaction of Keap-1 sequesters Nrf-2 in cytoplasm, leading to degradation by the proteasome, oxidation of Keap-1 at sulfhydryl groups, or phosphorylation of Nrf-2 results in a release of Nrf-2. Free of Keap-1, Nrf-2 translocates from cytoplasm to the nucleus and transactivates expression of antioxidants such as GSH, SOD, HO-1, GST and catalase [47]. In the present study, the IHC data demonstrated a decrease in the expression of Keap-1 and enhanced expression of HO-1 and Nrf-2 protein in the hippocampal region of Bergenin-treated animals compared to transgenic-saline-treated group. Western blot analysis further confirmed the antioxidant potential of Bergenin by significantly increasing Nrf-2 level in the hippocampus. Additionally, Bergenin was observed to significantly increase the antioxidant level in the HC, PFC and EC regions of mouse brains compared to the transgenic-saline-treated group. 

As a lipid-rich tissue with relatively low antioxidant potential makes it more susceptible towards oxidative stress, disturbing lipid metabolism [48]. The high content of polyunsaturated fatty acids is sensitive towards ROS, resulting in lipid peroxidation due to oxidative stress in the central nervous system (CNS). Myeloperoxidase, a heme protein expressed in phagocytic cells is a potent catalyst for the production of cytotoxic oxidants in human brain, increased expression of the enzyme has been shown in AD pathology [49]. MPO oxidizes nitrite to reactive nitrogen species and initiates lipid peroxidation. Treatment with Bergenin significantly decreased oxidative stress level in PFC, EC, and hippocampal region in comparison to the negative control group. Furthermore, NO level in plasma and various regions of interest in mouse brains were remarkably reduced in the Bergenin-treated mice compared to 5xFAD Tg mice.

Continuous aggregation and elevation of Aβ might contribute towards the innate immune response by activating microglia, resulting in neuronal loss via phagocytosis. The Aβ aggregation may activate immunological receptors such as TLR-4, resulting in the activation of NF-κB signaling pathway producing neurotoxic mediators, causing neuronal cell death [9]. The IHC findings in the present study demonstrated decreased expression of TLR-4 and NF-κB proteins in the hippocampal region of mice treated with Bergenin compared to transgenic-saline control animals. Further western blot analysis also confirmed the neuroprotective effect of Bergenin displaying decrease in NF-κB, while maintaining IκB-α level in hippocampus. In addition, Aβ aggregation produces inflammatory responses, releases inflammation-related mediators such as pro-inflammatory cytokines [50]. The treatment with Bergenin (60 mg/kg) significantly decreased pro-inflammatory cytokines (TNF-α and IL-1β) production compared to the 5xFAD Tg mice. 

Excessive generation of free radicals (ROS) are found to be associated with neurodegeneration [51]. The oxidized proteins and peroxidized lipids generates toxic products that migrate to different parts of neurons, impairs membrane integrity and cellular activity, thus increasing neuronal death [52]. An imbalance in apoptotic molecular markers results in neuronal apoptosis involved in neurodegeneration. The immunohistochemical data demonstrated that Bergenin promoted cell survival by increasing the expression of Bcl-2, an anti-apoptotic protein, while it downregulated the expression of pro-apoptotic Bax protein in the hippocampal regions of the mouse brains in comparison to the transgenic mice expressing increased pro-apoptotic and decreased anti-apoptotic proteins. Caspase-3 activated by Aβ accumulation aids in the pathophysiology of AD [52]. Moreover, Bergenin has remarkably decreased the expression of cleaved Caspase-3 in hippocampal neuronal cells compared to the Tg-saline group. Furthermore, the results from PI/Annexin-V staining showed a decrease in the early and late apoptosis, reducing overall apoptosis in the hippocampal neuronal cells compared to the transgenic-saline group.

The progress of AD is linked to the conformational changes and misfolding of Aβ peptides, leading to aggregation into oligomers and mature fibrils [53]. These extracellular plaques are mostly deposited in brain regions, such as the entorhinal cortex, hippocampus and prefrontal cortex, influencing memory, learning, and emotional behaviors. The histological analysis in this study suggested a significant decrease in neuronal damage in dentate gyrus of Tg treated with Bergenin. Similarly, it prevented degenerative changes in the neurons of the prefrontal cortex compared to the 5xFAD Tg group. The morphological and functional changes associated with AD could be considered as a result of the appearance of pathological granular structures in the hippocampus and cortical (PFC and EC) regions of Tg mouse brains [53]. The histopathological examination further confirmed a decrease in amyloid load in the Bergenin-treated transgenic mice in comparison to the negative control group in all the regions of interest. PAS granules were found in clusters in the brain parenchyma of several animal species, located in the CA1 region of the hippocampus extending gradually to other hippocampal layers and regions [54]. The appearance of these PAS granules can be related to the increased oxidative stress resulting in the modification of proteins and lipids [54]. Numerous studies have suggested the involvement of these granule formations as a protective strategy of neurons extruding damaged or misfolded proteins, which are subsequently engulfed by glia [55]. These round to ovoid clusters of granules were stained PAS-positive for the negative control group compared to the wild-type. However, Bergenin significantly decreased the PAS-positive granules near hippocampal and cortical regions of transgenic mice. 

Molecular docking analysis of Bergenin was performed to determine the plausible binding interaction within the active sites of various target proteins, i.e., Keap-1, Nrf-2, HO-1, TLR-4, IκB-α, NF-κB, Bcl-2, Bax, Caspase-3, and Aβ proteins. The ligand (Bergenin) and proteins were modified using MGL tools autodoc vina to remove any chemical entity that could hinder ligand-protein interaction. To describe the interaction of Bergenin with various proteins at their active site, the binding affinities were determined and represented in the heatmap. The lower values obtained demonstrated more stable complex formation between the target protein and ligand. The binding interaction of Bergenin and these proteins displayed H-bonding interactions revealing that the compound was efficiently docked with the target structures at their respective active site. 

In summary, results obtained from the in vitro, in vivo, and in-silico analysis suggested a neuroprotective effect of Bergenin. It showed no cytotoxicity and exhibited an antioxidant effect against H_2_O_2_-induced oxidative stress in HT-22 and PC-12 cells. Bergenin significantly attenuated spatial memory deficit in 5xFAD Tg mice by improving all the behavior, spectroscopic, pathological and biochemical parameters via regulation of Nrf-2/NF-κB/Bcl-2 and Reelin signaling pathway. 

## 4. Materials and Methods

### 4.1. Chemicals and Reagents

Fetal bovine serum (FBS), Dulbecco’s Modified Eagle Medium (DMEM), streptomycin, and penicillin were purchased from Sigma-Aldrich (St. Louis, MO, USA). MTT (3-[4,5-dimethylthiazol-2-yl]-2,5-diphenyl tetrazolium bromide), dimethyl sulfoxide (DMSO), diaminobenzidine substrate (DAB), Superoxide dismutase (SOD), reduced glutathione (GSH), Hydrogen peroxide (H_2_O_2_), Thiobarbituric acid (TBA), ascorbic acid, trichloroacetic acid, Griess reagents, and hexadecyltrimethylammonium bromide (HTAB) were purchased from Sigma-Aldrich (St. Louis, MO, USA). Propidium iodide (PI) and annexin-V-FITC were obtained from BD Biosciences (San Diego, CA, USA). The primary antibodies such as anti-Reelin, anti-Dab-1, anti-Aβ (1-42), anti-cleaved caspase-3, anti-Bcl-2, anti-Bax, anti-TLR-4, anti-NF-κB, anti-Keap1, anti-Nrf-2 and anti-HO-1 antibodies were obtained from Santa Cruz Biotechnology (CA, USA). For western blot antibodies such as Nrf-2 (Cat# sc-365949), NF-κB (Cat# sc-7386), IκB-α (sc-1643), Bcl-2 (sc-7382), and β-actin (Cat# sc-47778) were obtained from Santa Cruz Biotechnology (CA, USA).

### 4.2. Plant Material

Bergenin was isolated from the rhizome of *Bergenia ciliata* and was identified by our group as per patent application number 596/2019 (Figure 1A). Rhizomes of *Bergenia ciliata* were collected in June 2018 from Nathiagali, district Abbottabad, Khyber Pakhtunkhwa (KPK), Pakistan. After cleaning and thorough drying, the plant material was comminuted or crushed to a coarse powder. It was macerated with methanol (1:4) thrice and the filtrate was evaporated to get the dry extract. Dried extract was suspended in deionized water and extracted through solvent-solvent extraction by ethyl acetate and n-butanol, respectively. Organic solvent was used in an equal volume of the suspension and was extracted thrice; precipitates in the aqueous layer were collected and dissolved completely in methanol to crystallize Bergenin. Raw crystals of Bergenin were again recrystallized in methanol to get high purity (99.5%) Bergenin, analyzed by HPLC. 

### 4.3. Cells and Culture Media

The neuronal HT-22 cells and PC-12 cells were obtained from American Type Culture Collection (ATCC) (Manassas, VA). HT-22 and PC-12 cells were cultured in Dulbecco’s modified Eagle’s medium (DMEM) supplemented with 10% fetal bovine serum (FBS). These cells were maintained in the presence of 100 µg/mL streptomycin and 100 U/mL penicillin at 37 °C in a humidified atmosphere containing 5% CO_2_ and 95% air. Stock solution of 100 mM was prepared by dissolving the sample (Bergenin) in dimethyl sulfoxide (DMSO). Which was diluted further for working concentrations, i.e., 1, 10, 50 and 100 µM of Bergenin, keeping the final concentration of DMSO was < 0.2% to avoid any interference with the assay. Upon achieving the cell density of 70–80%, HT-22 and PC-12 cells were exposed to the indicated concentration of Bergenin for 2 h. The cells were then treated with H_2_O_2_ for 24 h. Then various assays were performed. 

### 4.4. Cell Viability Assay 

The cytoprotective effect of Bergenin on H_2_O_2_-induced HT-22 and PC-12 cells was measured by an MTT-based cell viability assay [20,56]. Briefly, HT-22 and PC-12 cells were plated at a density of 1 × 105 cells per well in a 24-well plate and were incubated for 24 h at 37 °C. The cells were treated with 1, 10, 50 and 100 µM of Bergenin and vehicle alone for 2 h prior to H_2_O_2_ (200 µM) induction and then were incubated for an additional 24 h at 37 °C. MTT solution at a final concentration of 0.5 mg/mL was added into the cell culture media and was incubated for 2 h at 37 °C in the dark. Finally, after the removal of supernatants from each well for further assays, DMSO was added to solubilize the formazan crystals. Absorbance was noted after 10 min at 595 nm with a microplate reader. The relative cell viability was calculated and compared with the untreated control group. 

### 4.5. In-Vitro Determination of Nitric Oxide (NO) Production 

After 24 h of incubation, nitric oxide (NO) assay was performed in HT-22 and PC-12 cells according to the Griess reaction method [57]. An amount of 100 mL of cell-free supernatant was collected from each well plate, and 100 mL Griess reagent was added. Curcumin (10 µM) was used as a positive control. The standard curve was calculated by using known concentrations of sodium nitrite, and the absorbance was measured at 540 nm using a microplate reader. 

### 4.6. In-Vitro Determination of Antioxidants and Oxidative Stress Level

The HT-22 and PC-12 cells were seeded in 24 well plates. The cells were pretreated with Bergenin (1, 10, 50 and 100 µM) and curcumin (10 µM) for 2 h accompanied with 200 µM H_2_O_2_ for another 24 h. The cells were then harvested to measure the intracellular antioxidant and oxidative stress levels. The reduced glutathione level (GSH) in HT-22 and PC-12 cells was investigated based on the reaction between reduced GSH and 5′,5′-dithiobis-2-nitrobenzoic acid (DTNB). The absorbance was measured at 412 nm. Additionally, superoxide dismutase (SOD) in HT-22 cells was determined by measuring the enzyme activity that inhibits auto-oxidation of pyrogallol by 50%. The enzyme activity was measured at 420 nm, expressed in percentage compared to the negative control group. Finally, the myeloperoxidase (MPO) level was determined by o-dianisidine method as reported previously. The enzyme activity was measured through microplate reader at 540 nm; the results were expressed as a percentage.

### 4.7. Animals

The B6SJL-Tg mice model of Alzheimer’s disease, sub-strain of JAX^®^Strain 006554, MMRRC034840 (APPSwFlLon,PSEN1*M146L*L286V) 6799Vas/Mm, referred to as 5xFAD transgenic mice, were purchased from Jackson Laboratory (Sacramento, CA, USA). Non-transgenic wild-type (WT) animals were used in all experiments as normal/vehicle control. The animals used in the experiment were mixed-breed consisting of both male and female mice. Animals were kept in the animal facility of Department of Pharmacy, University of Peshawar (UOP), Khyber Pakhtunkhwa (KPK), Pakistan. They were housed at 25 ± 2 °C temperature with 60 ± 10% relative humidity and a 12-h light/dark cycle. Routinely beddings were changed in all the cages. All mice had complete access to standard chow and water ad libitum. All the experiments were approved by the Bioethical Committee of UOP with an approval No. 12/EC-17/Pharm. The study was conducted in a pathogen-free facility in strict accordance with the guidelines. 

### 4.8. Experimental Groups 

The animals were divided into six groups. Each group consisting of ten animals (*n* = 10).

Group I: Wild type or non-transgenic mice, Group-II: Negative control group (5xFAD Tg-saline), Group-III: Positive group (5xFAD Tg-galanthamine, 8 mg/kg, i.p.), Group-IV: Treatment-I (5xFAD Tg-Bergenin, 1 mg/kg, oral), Group V: Treatment-II (5xFAD Tg-Bergenin, 30 mg/kg, oral), Group VI: Treatment-III (5xFAD Tg-Bergenin, 60 mg/kg, oral).

### 4.9. Dose Preparation and Administration

A fresh solution of Bergenin was prepared at a concentration of 1, 30, and 60 mg/kg in normal saline (N.S) and dimethyl sulfoxide (DMSO) according to the required volume for oral administration, i.e., 250 µL/mouse/day. A standard anti-cholinesterase, i.e., galanthamine, was selected as a positive control at a dose of 8 mg/kg [58]. Normal saline was administered to wild-type/non-transgenic mice and transgenic (5xFAD Tg) mice.

### 4.10. Experimental Design and Sample Collection

All the experiments were performed in a well-illuminated room with 20–22 °C temperature. The schematic diagram representing the experimental design and sample collection is shown in Figure 15. Behavioral assessment was done to evaluate the escape latency in the Y-Maze test. The training session was conducted for nine consecutive days and test trials were conducted 1 h after drug administration and escape latency was recorded. On the 14th day, percent spontaneous alteration behavior (SAB) (%) was observed after drug administration. Animals were sacrificed by cervical dislocation and blood was collected by cardiac puncture. The mice brain were quickly dissected and sectioned into HC, PFC and EC regions, respectively. Some of the brain tissues were fixed in 4% paraformaldehyde for histological analysis, while others were homogenized at 4 °C in 0.1 M phosphate buffer (10% *w*/*v*, pH 7.4) and were stored at −80 °C after centrifugation for further neurochemical analysis. 

### 4.11. Preparation of Brain Tissues for Histological Analysis

Animals were anesthetized and transcardially perfused with phosphate-buffered saline (PBS) containing 4% paraformaldehyde [59]. Brains were immediately removed and sectioned coronally at 4 µm using a microtome, postfixed in same fixative for 12 h and paraffin-embedded. The brain tissues were sectioned using a microtome and paraffin-embedded sections were mounted on glass slides for further analysis.

### 4.12. Behavioral Analysis

#### 4.12.1. Y-Maze Test

The Y-Maze apparatus was used for the determination of spatial memory in experimental animals [58,60]. The apparatus was made of polystyrene with three arms perpendicular to each other, having a dimension of 30 cm × 8 cm × 20 cm and three true-false exits, i.e., one true and two false exit arms. A removable pipe was attached to the true exit for the safe transfer of mice to their respective cage. The water level was maintained at 2 cm with a temperature of 20–25 °C, in order to provide easy escape and smooth padding to the animals. All the experimental mice were trained in the Y-Maze apparatus for nine consecutive days, three trials each for 60 s with an interval of 1 h. Each mouse was placed at the end of the closed arm facing opposite to the center of the apparatus. A semi-random sequence was followed in selection of the starting arm, with three successive trails made at the same point. The animals which failed to escape within 60 s were manually guided to the exit hole with the help of transparent Plexiglas slide. From 10th day onwards to the 14th day, i.e., five consecutive days test trials were performed. Wild-type or non-transgenic mice and 5xFAD transgenic mice received saline and the treatment groups received Bergenin (1, 30, and 60 mg/kg, orally). Post 1-h of drug administration, mice were subjected to Y-Maze test and escape latency for each mouse was recorded. On the last day of drug administration, spontaneous alteration behavior (SAB) was observed by placing the individual mice in the center of the maze and allowing their free movement for 8 min. The number of entries in each arm was recorded for three consecutive sessions. SAB was defined as the consecutive entries into three arms in an overlapping triplet sets. The percent (%) alternation behavior was calculated as follows: [successive triplet sets (entries into three different arms consecutively)/total number of arm entries-2] × 100.

#### 4.12.2. Morris Water Maze (MWM) Test

The effect of the Bergenin on the spatial learning and memory of 5xFAD Tg mice was evaluated through the Morris water maze apparatus (MWM, diameter: 110 cm, height: 40 cm) containing opaque water (depth: 20 cm) maintained at 22–23 °C, following previously reported method [61]. The maze consisted of four quadrants with a hidden platform (diameter: 7 cm, 1 cm below the water surface) in one of the four quadrants. Animals were trained for 5 days with four trials per day, with an interval of 1 h. Each trial began by placing the mouse into a different quadrant and allowing it to swim freely for a maximum of 60 s and the escape latency time in seconds was determined to find the submerged platform. The animals which failed to find the platform in one minute, were guided manually, and allowed to stay for 10 s on the platform. After training, on the sixth day, the memory retention of the mice was tested in the probe trial (duration 60 s). The time spent by mice in the target quadrant was recorded. 

#### 4.12.3. Open Field Test (OFT) 

A square, open field having 50 cm × 50 cm dimensions divided into 16 squares, surrounded by 50-cm-high wooden walls was used for the analysis of locomotor activity and anxiety-like behavior in 5xFAD Tg mice. The tests were carried out in a soundproof room, lit by a 60-W light bulb about 1.75 m above the center of the open-field with an illuminance of 150 lux at the center of the apparatus [62]. At the start of the test, animals were placed in the central zone and parameters like exploratory (locomotor) activity, number of rearing, and time spent in the center were recorded by a video camera for 5 min. The locomotor activity was estimated by a total number of crossings; a crossing was considered when the animal would enter a square with both of its hind paws. A rearing episode was recorded, whenever the mice stood on their hind paws with an angle greater than 45° between its body and the floor. Additionally, the time spent in the center of the open field test was also evaluated [63]. 

### 4.13. Biochemical Analysis

#### 4.13.1. Fourier Transform-Infrared (FT-IR) Analysis

The FT-IR spectrometry was performed to evaluate the protein and lipid content in brain tissues. Additionally, β-sheet secondary structure was observed to determine the amyloid fibril formation [39,64]. The brain tissues were lyophilized for 12 h in a lyophilizer (Alpha 1-2 LDplus, Martin Christ, freeze dryer) before subjected to the FT-IR spectrometer. The absorbance spectra were recorded in the 4000–500 cm^−1^ regions at room temperature (25 ± 1 °C) on an FT-IR spectrometer IR tracer (Shimadzu, Japan). The secondary derivatives of the FT-IR spectra were calculated by increasing the resolution of the obtained FT-IR spectra. The spectra were analyzed using Origin 8.5 software. The absorption bands for lipids, proteins, phospholipids and nucleic acids in Table 4 were defined according to the literature [30,65,66].

#### 4.13.2. Differential Scanning Calorimeter (DSC)

The DSC measurements were obtained using a Nano DSC instrument (Thermal Analysis Instruments-Waters LLC). Typically, the temperature was increased from 25–140 °C, with 2–4 heating and cooling scans obtained at heat rates of 1.5 °C/min [67]. Thermal denaturational profile was displayed by the first heating scan, while the successive heating scans demonstrated same profiles with no detectable thermal events, representing denatured samples. 

#### 4.13.3. Evaluation of Histopathological Changes in Hematoxylin & Eosin (H&E) Stained Tissues

Histological alterations in the cytoarchitecture of the brain tissues were identified by staining the brain sections with hematoxylin & eosin (H&E) [68]. The degenerative changes in the granular layer of the hippocampus and prefrontal cortex were observed in each experimental group. The stained tissues were examined under the light microscope using 50 µm scale bars. Quantification was done with ImageJ software in different ROI, respectively. 

#### 4.13.4. Assessment of Amyloid Beta Deposits in Brain Tissue

Amyloid deposits were quantified using Congo red-stained brain tissue [69]. The paraffin-embedded slides were de-waxed using ethanol and xylene solution. The saturated solution of Congo red and sodium chloride solution were prepared to stain Aβ deposits in mice brain tissue. Light microscope was used to observe the presence and location of Aβ deposit associated with memory impairment in AD. Evaluation of amyloid loads was performed in different regions of interest (ROI), i.e., the prefrontal cortex, hippocampus and entorhinal cortex of the mice brains. Images were then processed by using ImageJ software 1.48 version (NIH, Rockville, MD, USA).

#### 4.13.5. Assessment of Periodic Acid-Schiff (PAS) Staining in Brain Tissue

Each mouse brain section was carefully mounted on the glass slide and was stained with periodic acid-Schiff (PAS). The pathological granular structures in different regions of interest (ROI), i.e., the hippocampus, prefrontal cortex and entorhinal cortexes of the mouse brains were stained positively with periodic acid-Schiff (PAS) [70,71]. The granules were observed to be round to ovoid in appearance, when observed under light microscope using scale bars 50 µm. The evaluation of PAS-positive granules in different ROI was performed using ImageJ software.

#### 4.13.6. Western Blot Analysis

The western blot analysis were performed as described previously [72]. The hippocampal tissues were homogenized with a homogenizer (Polytron PT-10-35-GT, Thomas Scientific, Swedesboro, NJ, USA) in an ice bath. The tissue homogenates were then quantified using the Bio-Rad Protein Assay according to the manufacturer’s protocol. Then, 20–40 μg of total protein for IκB-α, NF-κB, Nrf-2, and Bcl-2 were prepared in sodium dodecyl sulfate (SDS) sample buffer consisting of 5% β-mercaptoethanol, 2% SDS, 10% glycerol, and 60 mM Tris-HCl (pH 6.8). Then IκB-α, NF-κB, Nrf-2, Bcl-2, and β-actin were separated on a 10–12% SDS-PAGE gel, and were shifted onto a polyvinylidene fluoride (PVDF) membrane (Amersham Protran, Sigma-Aldrich, St. Louis, MO, USA), blocked with 3% albumin solution and blotted with primary antibodies (1:1000) and their corresponding secondary antibodies (1:5000). For the detection of antibodies, chemiluminescent signals were developed by Clarity ECL Western Blotting Substrate (Bio-Rad) according to the manufacturer’s instructions. For the quantification of target bands, UN-SCAN-ITTM software version 6.1 (Silk Scientific Co., Orem, UT, USA) was used. 

#### 4.13.7. Immunohistochemical Analysis

The immunohistochemical staining was performed to investigate the effect of Bergenin on Reelin signaling pathway (Reelin/Dab1) and amyloid plaques (Aβ1-42). In addition, effect of Bergenin was evaluated on inflammatory (TLR-4/NF-κB), antioxidant (Keap1/Nrf-2/HO-1) and apoptotic (Cleaved-casp-3/Bcl-2/Bax) proteins in the hippocampal regions of transgenic and wild-type mice [33,73]. For 3, 3′-diaminobenzidine (DAB) staining brain sections were incubated with primary antibodies, i.e., anti-Reelin (mouse polyclonal), anti-Dab1 (mouse polyclonal), anti-Aβ (1-42) (mouse polyclonal), anti-TLR-4 (mouse polyclonal), anti-IκB-α (rabbit polyclonal), anti-NF-κB (rabbit polyclonal), anti-Keap1 (rabbit polyclonal), anti-Nrf-2 (rabbit polyclonal), anti-HO-1 (rabbit polyclonal), anti-Bax (rabbit polyclonal), anti-cleaved caspase-3 (rabbit polyclonal), and anti-Bcl-2 (mouse polyclonal) overnight at 4 °C. Further, the sections were incubated with biotin-conjugated secondary antibodies for one hour at room temperature; avidin-biotin complex (ABC) kit and DAB kit (Vector Laboratories, Burlingame, CA, USA) were used to detect signals. The sections were observed using a light microscope at 100X. ImageJ software 1.48 version (NIH, Rockville, MD, USA) was used for the quantification of the images and were expressed as relative expression of the treated samples compared to the 5xFAD Tg control group.

#### 4.13.8. Detection of Apoptotic Cells by Annexin V Staining

Flow cytometry was performed to determine the apoptosis in the hippocampal neuronal cells by using Annexin V-Fluorescein isothiocyanate (FITC) Apoptosis Detection kit (Sigma-Aldrich), according to manufacturer’s instructions [74]. The hippocampal cells homogenate was diluted with chilled PBS and were loaded in the Percoll medium; the preparation was centrifuged at 1000 rpm for 30 min at 4 °C. The ring at the interface of Percoll separation was collected and washed twice with PBS. The pellets were resuspended and diluted in PBS; cells were counted on a haemocytometer using Trypan blue. Subsequently, 1 × 10^6^ cells were successively incubated with 500 µL binding buffer, 5 µL propidium iodide and 5 µL Annexin V-FITC for 15–20 min in a dark room. The samples were acquired immediately with flow cytometer (BD Biosciences, San Jose, CA, USA).

#### 4.13.9. Determination of Anti-Oxidant Level

The brain tissue homogenates were centrifuged at 1000 rpm for 10 min at 4 °C, the supernatant obtained was used further for the assay of reduced glutathione (GSH), superoxide dismutase (SOD), glutathione-S-transferase (GST), and catalase (CAT) activity.

Reduced glutathione (GSH) level: Reduced glutathione (GSH) level was determined in the brain tissue homogenate based on the development of yellow color when 5-5-dithiobis-2-nitrobenzoic acid (DTNB) reacts with sulfhydryl groups [75]. According to Ellman’s method, brain sections (HC, PFC and EC) were homogenized and a portion of homogenate was precipitated with 5% trichloroacetic acid (TCA), to separate proteins. The mixture was kept at room temperature for 15 min, centrifugation was done at 2500× *g* and the supernatant was stored at –80 °C. The reduced GSH level was determined by the reaction of freshly prepared supernatant DTNB in 0.2 M sodium phosphate buffer (pH 8.0) and the supernatant. The absorbance was read at 412 nm and the activity was expressed in percentage.

Superoxide dismutase (SOD) level: SOD activity was determined by using the method of Marklund et al. [76]. The reaction mixture was composed of 50 mM Tris-EDTA buffer (pH 8.5), 24 mM pyrogallol and 10 µL of the supernatant/sample in a final volume of 0.2 mL. The enzyme activity was calculated at 420 nm and was expressed in percentage.

Glutathione-S-transferase (GST) level: The GST activity was evaluated according to the previously reported method by Warholm et al. [77]. Change in the absorption was measured after the addition of 1-chloro-2, 4-dinitrobenzene (CDNB) in the reaction mixture containing glutathione as a co-substrate and the sample/supernatant recorded at 340 nm by spectrophotometer [73]. The results were expressed in percentage. 

Catalase level: Furthermore, CAT activity was obtained from the rate of H_2_O_2_ decomposition upon addition of sample/supernatant at 240 nm according to the method explained by Aebi et al. (1974) [78,79,80]. The enzyme activity was expressed in percentage.

#### 4.13.10. Evaluation of Oxidative Stress Level

Malondialdehyde (MDA) level: Malondialdehyde, a marker of LPO was detected by measuring thiobarbituric acid reactive substances (TBARS) in supernatants obtained from brain tissue homogenates (PFC, EC and HC) as previously described [81]. LPO indirectly measures oxidative stress by ROS. The LPO level was determined by measuring the quantity of MDA level in each sample; the absorbance was determined at a wavelength of 535 nm using a microplate reader. The enzyme activity was expressed in percentage.

Myeloperoxidase level: Various oxidants are generated by Myeloperoxidase (MPO) which contribute towards tissue damage during inflammation [49]. The MPO level in the brain tissue homogenates (HC, PFC and EC) were determined by adopting Hexadecyltrimethylammonium bromide (HTAB) buffer and o-dianisidine method as reported previously [82,83]. The brain tissues were homogenized in HTAB prepared in 50 mM phosphate buffer (pH 6.0), resulting in the release of MPO from the cells. The homogenized sample was frozen and thawed thrice before centrifugation. The mixture of o-dianisidine and hydrogen peroxide was added to the supernatant, and the MPO activity was determined at 540 nm using ELISA plate reader. 

#### 4.13.11. Determination of Nitrite Content 

Nitrite content was determined in PFC, EC and HC regions and plasma by using Griess reagent [84,85]. Equal quantities of supernatant and Griess reagent were incubated for 15 min at room temperature. Absorbance was measured at 560 nm on an ELISA plate reader and nitrite content was measured from the standard curve produced by sodium nitrite used as a standard, expressed in percentage. 

#### 4.13.12. Evaluation of Pro-Inflammatory Cytokines Level in Mice Brain Tissue

Pro-inflammatory cytokine production (IL-1β and TNF-α) were assessed in different ROI (PFC, HC and EC) regions of mouse brains by quantitative ELISA analysis (eBioscience, Inc., San Diego, CA, USA) using a microplate reader [86,87]. The cytokine level was estimated in the tissue from all the experimental groups, i.e., wild-type, 5xFAD Tg, and Bergenin-treated groups, following the manufacturer’s protocol. The concentration of cytokines in tissues was indicated in percentage compared to the control.

#### 4.13.13. Molecular Docking Analysis 

The structural activity relationship of Bergenin was explored by adopting molecular docking interaction with various receptor proteins by using Autodock Vina_4.2 along with discovery studio software as reported previously [88,89]. The 3D structure of TLR-4 (PDB ID: 3VQ2), IκB-α (PDB ID: 6Y1J), NF-κB (PDB ID:1VKX), Keap-1 (PDB ID: 1U6D), Nrf-2 (PDB ID: 2FLU), HO-1 (PDB ID: 1irm), Bcl-2 (PDB ID:2W3L), Bax (PDB ID:5W62), Caspase-3 (PDB ID:2XYG) and amyloid peptide (PDB ID: 4mvl) were downloaded from the protein data bank (pdb). All the water molecules and heteroatoms were removed, and polar hydrogens were added to the protein downloaded from the PDB. Similarly, the ligand, i.e., Bergenin was downloaded from PubChem and saved as PDB format. The discovery studio visualizer_2016 software was used to visualize the results. The lowest binding energy indicated the highest binding affinity [90]. In order to display the multi-target binding efficiency of the ligand, a heatmap was generated based on binding energy scores by using GraphPad Prism version 8.5.

#### 4.13.14. Statistical Analysis

The results of the study were expressed as mean ± standard deviation (S.D). Two-way analysis of variance (ANOVA) was applied followed by Bonferroni’s post hoc test for the assessment of statistical significance amongst various treated groups. GraphPad Prism version 8.5 was used for the analysis of the behavior and biochemical data, while, histological images were analyzed using ImageJ software 1.48 version (NIH, Rockville, Maryland, USA). The target bands of western blot were quantified using UN-SCAN-ITTM software Version 6.1 (Silk Scientific Co., Orem, UT, USA). For the statistical significance, the value of *p* < 0.05 was selected as the criterion of significance difference.

## 5. Conclusions

The present study demonstrated the neuroprotective effect of Bergenin against spatial memory impairment associated with AD (Figure 16). Preliminarily, Bergenin presented no cytotoxicity in HT-22 and PC-12 cells and produced an antioxidant effect. A significant improvement in pathological and biochemical parameters affecting memory in 5xFAD Tg mice was shown by treatment with Bergenin. Additionally, an appreciable restoration of biomolecules such as lipids, proteins and their derivatives was observed with Bergenin compared to 5xFAD Tg control group. The results obtained from western blot and IHC analysis suggested its anti-inflammatory, antioxidant and antiapoptotic effect against spatial memory deficit in 5xFAD Tg mice. Bergenin was also observed to regulate Reelin signaling pathway by significantly enhancing Reelin expression and thus decreasing Aβ aggregation in the hippocampal regions of the mouse brains. 

## Figures and Tables

**Figure 1 ijms-22-06603-f001:**
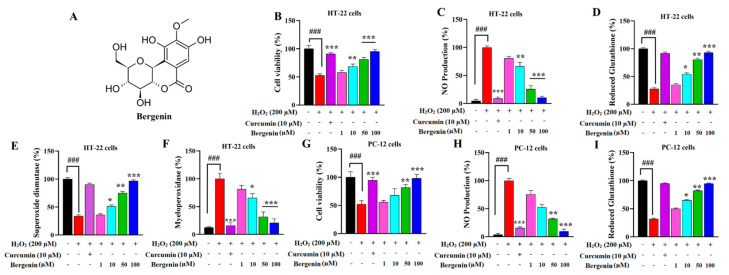
(**A**) Chemical structure of Bergenin. Effect of Bergenin on (**B**) (%) cell viability, (**C**) (%) nitrite production, (**D**) (%) reduced glutathione, (**E**) (%) superoxide dismutase and (**F**) (%) Myeloperoxidase level in HT-22 cells exposed to H_2_O_2_ (200 µM) for 24 h. Effect of Bergenin on (**G**) (%) cell viability, (**H**) (%) nitrite production, (**I**) (%) reduced glutathione in PC-12 cells exposed to H_2_O_2_ (200 µM) for 24 h. Results were expressed as mean ± S.D. * *p* < 0.05, ** *p* < 0.01 and *** *p* < 0.001 versus the cells exposed to H_2_O_2_ alone, ### *p* < 0.001 versus the control (no H_2_O_2_).

**Figure 2 ijms-22-06603-f002:**
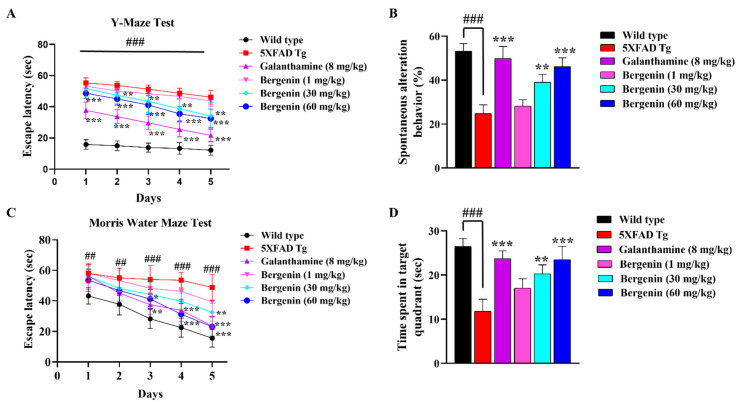
Bergenin attenuated spatial memory deficit in 5xFAD Tg mice. In the Y-maze test (**A**) Escape latency from day 1 to 5 and (**B**) % Spontaneous alteration behavior (SAB) on day 14th was evaluated in all the experimental groups. In the Morris water maze (MWM) test (**C**) Escape latency from day 1 to 5 and (**D**) Time spent in the target quadrant (probe test) on day 14th was evaluated in all the experimental groups. Results were expressed as mean ± S.D (*n* = 10). * *p* < 0.05, ** *p* < 0.01 and *** *p* < 0.001 versus the transgenic mice group, ## *p* < 0.01 and ### *p* < 0.001 versus the control (wild-type).

**Figure 3 ijms-22-06603-f003:**
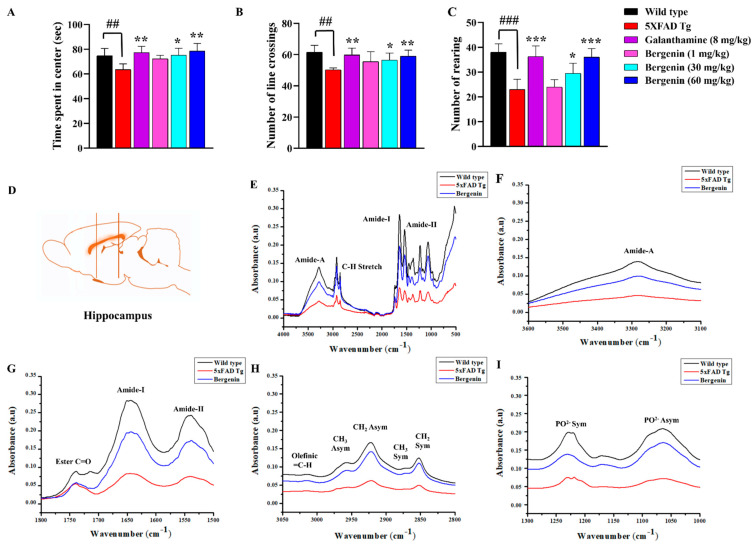
Bergenin attenuated anxiety-like behavior in 5xFAD Tg mice and exhibited no change in locomotor activity. (**A**) Time spent in center (s), (**B**) Number of line crossings, and (**C**) Number of rearing were evaluated in the open field test (OFT) for all the experimental groups. Results were expressed as mean ± S.D (*n* = 10). * *p* < 0.05, ** *p* < 0.01 and *** *p* < 0.001 versus the 5xFAD Tg mice group, ## *p* < 0.01 and ### *p* < 0.001 versus the control (wild-type). (**D**) Diagrammatic representation of the hippocampal (HC) region in the mouse brain. FT-IR spectra of the HC region of mice brain in wild-type, 5xFAD Tg mice, and Bergenin groups in (**E**) 4000–500 cm^−1^ (Full spectra), (**F**) 3600–3100 cm^−1^ (Amide-A region), (**G**) 1800–1500 cm^−1^ (Ester C=O, Amide-I, and Amide-II region), (**H**) 3050–2800 cm^−1^ (olefinic =C-H, symmetric and asymmetric CH3, and CH2 regions), and (**I**) 1300–1000 cm^−1^ (symmetric and asymmetric PO2- region).

**Figure 4 ijms-22-06603-f004:**
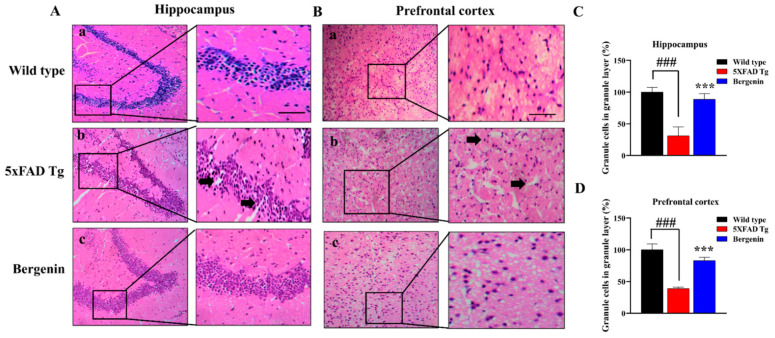
Representative photomicrographs of hematoxylin and eosin (H&E) stained hippocampal and prefrontal cortex region in (**a**) wild-type, (**b**) 5xFAD Tg, and (c) Bergenin groups. (**A**) Bergenin (60 mg/kg) displayed a regular density of granule cells with well-stained nuclei (arrows) in the hippocampal region while: (**B**) it prevented neuronal damage and poorly stained nuclei (arrows) in the prefrontal cortex. Scale bar = 50 µm. Quantitative analysis was done in (**C**) hippocampal and (**D**) prefrontal cortex region, results were expressed as mean ± S.D. *** *p* < 0.001 versus the 5xFAD Tg group, ### *p* < 0.001 versus the wild-type group.

**Figure 5 ijms-22-06603-f005:**
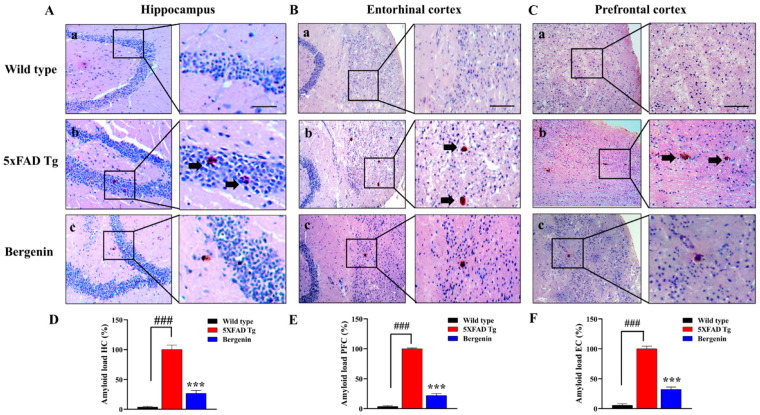
Bergenin (60 mg/kg) demonstrated a decrease in amyloid load (arrows) in all the regions of interest. Representative photomicrographs of dentate gyrus region of the hippocampus, entorhinal cortex, and prefrontal cortex in (**a**) wild-type, (**b**) 5xFAD Tg, and (**c**) Bergenin-treated groups stained with Congo red. Scale bar = 50 µm. (**D**–**F**) Quantitative analysis was performed, results were expressed as mean ± S.D (*n* = 3). *** *p* < 0.001 versus the 5xFAD Tg group, ### *p* < 0.001 versus the wild-type group.

**Figure 6 ijms-22-06603-f006:**
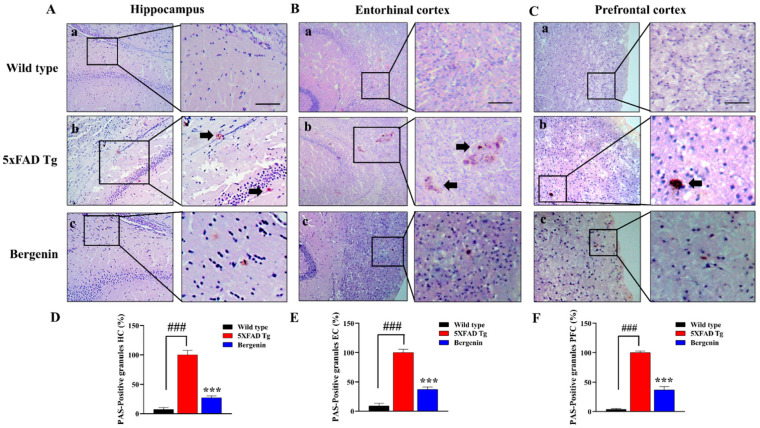
Bergenin (60 mg/kg) demonstrated a decrease in PAS-positive granules (arrows) in all the regions of interest. Representative photomicrographs of (**A**) hippocampal CA2, (**B**) prefrontal cortex, and (**C**) entorhinal cortex region in (**a**) wild-type, (**b**) 5xFAD Tg, and (**c**) Bergenin groups were stained with Periodic acid-Schiff (PAS) in 5xFAD Tg mice. Scale bar = 50 µm. (**D**–**F**) Quantitative analysis results were expressed as mean ± S.D. *** *p* < 0.001 versus the 5xFAD Tg group, ### *p* < 0.001 versus the wild-type group.

**Figure 7 ijms-22-06603-f007:**
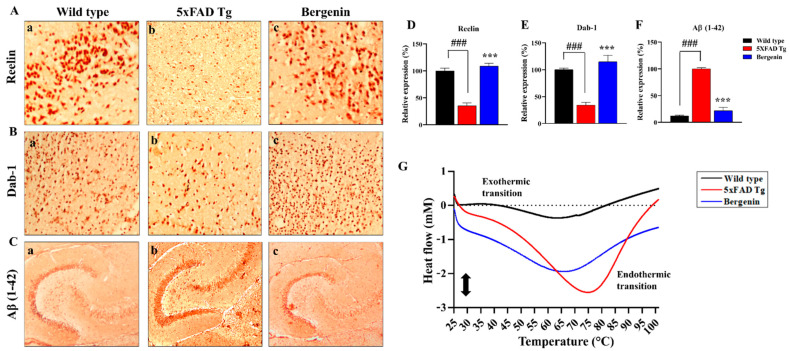
Bergenin demonstrated the protective effect through the Reelin-signaling pathway by inhibiting β-aggregation. Immunohistochemical analysis of (**A**) Reelin, (**B**) Dab-1 and (**C**) Aβ (1–42) in the hippocampal region of (**a**) wild-type, (**b**) 5xFAD Tg, and (**c**) Bergenin-treated groups. (**D**–**F**) Represents quantitative analysis of (%) relative expression of proteins. Scale bar = 50 µm. Results were expressed as mean ± S.D. *** *p* < 0.001 versus the 5xFAD Tg group, ### *p* < 0.001 versus the wild-type group. (**G**) DSC heating scans (25–100 °C) displaying thermal transitions in hippocampal regions of the mouse brains in wild-type, 5xFAD Tg, and Bergenin groups.

**Figure 8 ijms-22-06603-f008:**
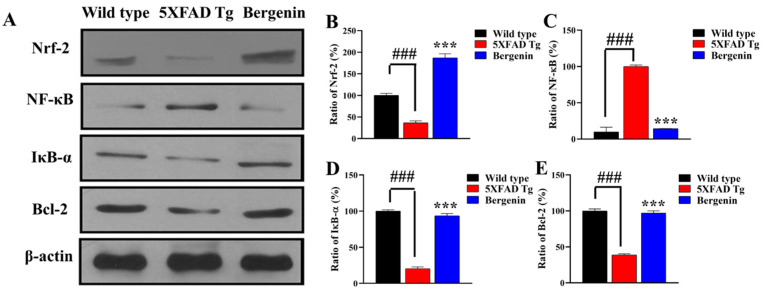
(**A**) Western blot analysis in the hippocampal regions of mouse brains for Nrf-2, NF-κB, IκB-α, and Bcl-2 proteins in wild-type, 5xFAD Tg and Bergenin-treated groups. (**B**–**E**) Represents quantitative analysis of target proteins expressing the ratio of percentage (%) production. Scale bar = 50 µm. Results were expressed as mean ± S.D. *** *p* < 0.001 versus the 5xFAD Tg group, ### *p* < 0.001 versus the wild-type group.

**Figure 9 ijms-22-06603-f009:**
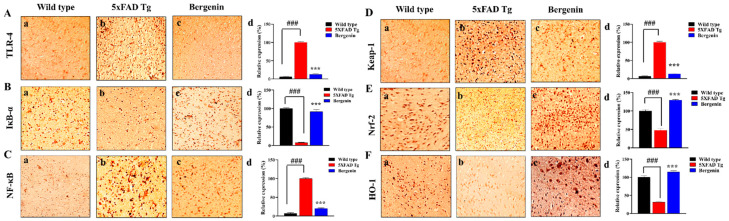
Immunohistochemical (IHC) analysis of (**A**) TLR-4, (**B**) IκB-α and (**C**) NF-κB proteins in the hippocampal region of (**a**) wild-type, (**b**) 5xFAD Tg, and (**c**) Bergenin-treated groups, (**d**) quantitative analysis of (%) relative expression of proteins. Immunohistochemical analysis of (**D**) Keap-1, (**E**) Nrf-2, and (**F**) HO-1 in hippocampal regions of mouse brains in (**a**) wild-type, (**b**) 5xFAD Tg, and (**c**) Bergenin-treated groups. (**d**) Represents the quantitative analysis of (%) relative expression of proteins. Scale bar = 50 µm. Results were expressed as mean ± S.D. *** *p* < 0.001 versus the 5xFAD Tg mice group, ### *p* < 0.001 versus the wild type.

**Figure 10 ijms-22-06603-f010:**
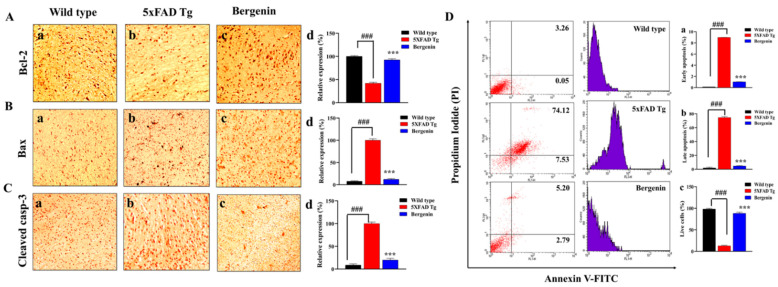
Immunohistochemical analysis of (**A**) Bcl-2, (**B**) Bax and (**C**) Cleaved caspase-3 proteins in hippocampal regions of the brain in (**a**) wild-type, (**b**) 5xFAD Tg, and (**c**) Bergenin-treated groups, where (**d**) represents a quantitative analysis of (%) relative expression of proteins. (**B**) Flow cytometry representing Annexin V/FITC and propidium iodide staining in hippocampal neuronal cells in (**a**) wild-type, (**b**) 5xFAD Tg, and (**c**) Bergenin-treated groups. Quantitative analysis were represented by (**a**) (%) early apoptosis, (**b**) (%) late apoptosis, and (**c**) (%) live cells. Scale bar = 50 µm. Results were expressed as mean ± S.D. *** *p* < 0.001 versus the 5xFAD Tg mice group, ### *p* < 0.001 versus the wild type.

**Figure 11 ijms-22-06603-f011:**
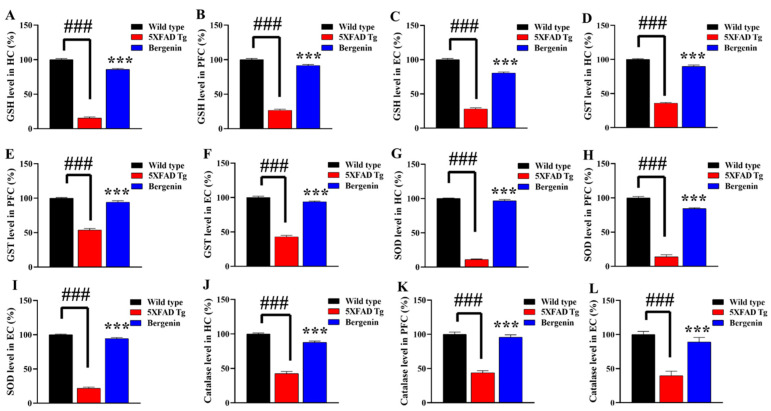
Bergenin (60 mg/kg) enhances antioxidant levels in transgenic mice. Antioxidant proteins and enzymes were determined in the hippocampal (HC), prefrontal cortex (PFC) and entorhinal cortex (EC) regions of mouse brains. (**A**–**C**) Reduced glutathione (%), (**D**–**F**) Glutathione-S-transferase (%), (**G**–**I**) Superoxide dismutase (%) (**J**–**L**) Catalase (%), level in different regions of mouse brains. Results were expressed as mean ± S.D. *** *p* < 0.001 versus the 5xFAD Tg mice group, ### *p* < 0.001 versus the wild type.

**Figure 12 ijms-22-06603-f012:**
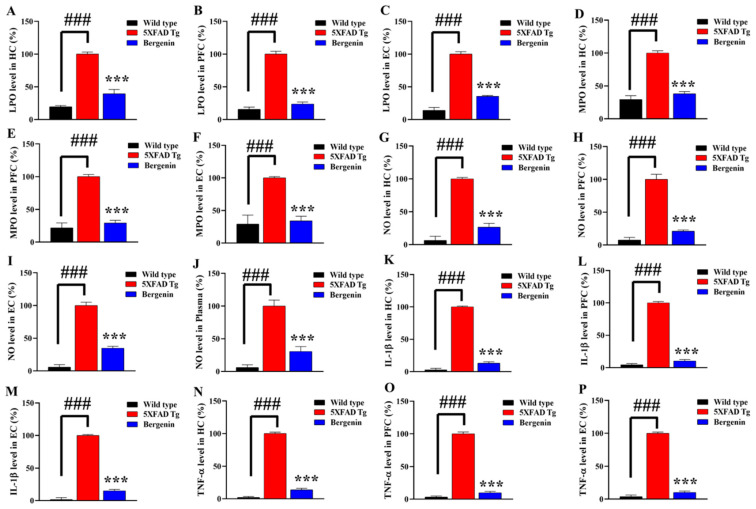
Bergenin ameliorates oxidative stress by decreasing (**A**–**C**) Lipid peroxidase (%) and (**D**–**F**) Myeloperoxidase (%) level in HC, PFC and EC regions of the mouse’s brain. Similarly, % Nitrite production in (**G**) HC, (**H**) PFC, (**I**) EC regions and (**J**) plasma of the mice were observed to be decreased with Bergenin treatment. Bergenin (60 mg/kg) displayed an inhibitory effect on pro-inflammatory cytokines (**K**–**M**) IL-1β, and (**N**–**P**) TNF-α production in HC, PFC and EC regions of mouse brains in wild-type, 5xFAD Tg and Bergenin groups. Results were expressed as mean ± S.D. *** *p* < 0.001 versus the transgenic (5xFAD Tg) mice group, ### *p* < 0.001 versus the wild type.

**Figure 13 ijms-22-06603-f013:**
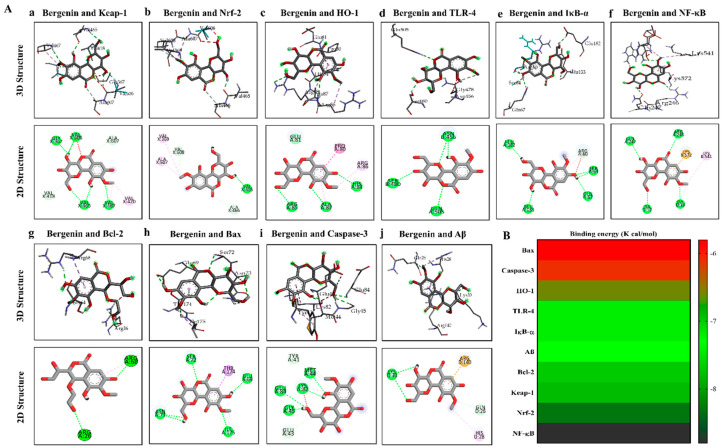
Molecular docking analysis of Bergenin with the active site of protein targets. (**A**) Three dimensional (3D) and (**B**) Two dimensional (2D) structural arrangement (**a**) Kelch-like ECH-associated protein (Keap1), (**b**) Nuclear factor erythroid 2-related factor 2 (Nrf-2), (**c**) Heme oxygenase (HO-1), (**d**) Toll-like receptors (TLR-4), (**e**) nuclear factor of kappa B inhibitor, alpha (IκB-α), (**f**) Nuclear factor kappa B (NF-κB), (**g**) B-cell lymphoma 2 (Bcl-2), (**h**) Bcl-2 associated X protein (Bax), (**i**) Caspase-3 (Casp-3), and (**j**) Amyloid beta (Aβ). (**B**) Binding energy (K cal/mol) scores were presented by Heatmap with multiple targets, i.e., Bax, Caspase-3, IκB-α, HO-1, TLR-4, Bcl-2, Keap-1, NF-κB, Nrf-2 and Aβ.

**Figure 14 ijms-22-06603-f014:**
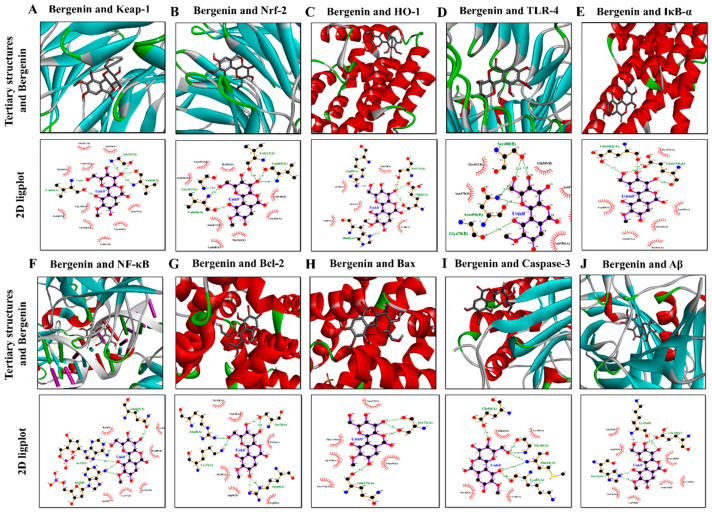
Representation of 3D (three dimensional) tertiary structure of protein with ligand and 2D (two dimensional) Ligplot structural interaction of ligand (Bergenin) with target proteins, i.e., (**A**) Keap1, (**B**) Nrf-2, (**C**) HO-1, (**D**) TLR-4, (**E**) IκB-α, (**F**) NF-κB, (**G**) Bcl-2, (**H**) Bax, (**I**) Casp-3 and (**J**) Aβ.

**Figure 15 ijms-22-06603-f015:**
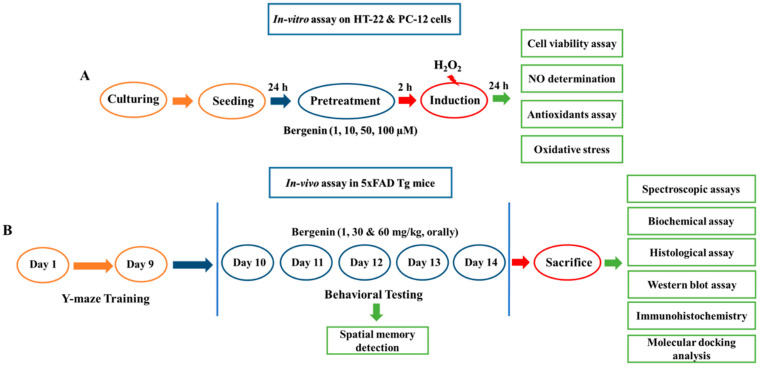
Schematic diagram representing the experimental design. (**A**) Preliminarily in-vitro assay was performed on HT-22 and PC-12 cells. (**B**) The in vivo assay was performed on 5xFAD Tg mouse model.

**Figure 16 ijms-22-06603-f016:**
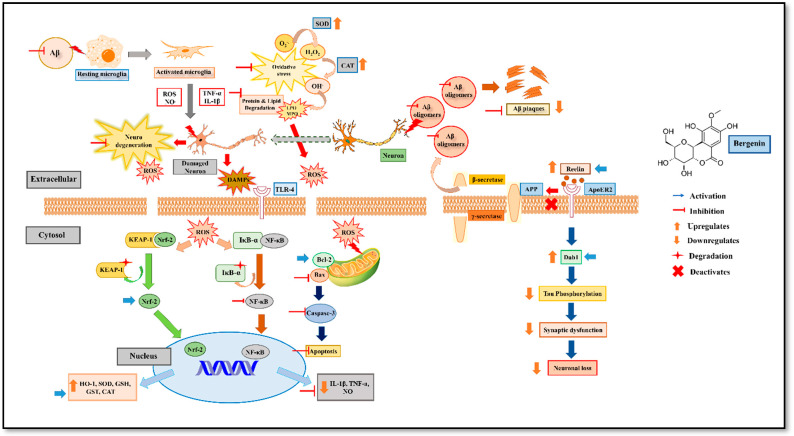
Graphical abstract representing the neuroprotective effect of Bergenin against spatial memory in 5xFAD Tg mice. Amyloid beta (Aβ), Amyloid precursor proteins (APP), Apolipoprotein E Receptor 2 (ApoER2), B-cell lymphoma 2 (Bcl-2), Bcl-2 associated X protein (Bax), Catalase (CAT), Disabled-1 adaptor protein (Dab 1), Damage associated molecular patterns (DAMPs), Glutathione (GSH), Glutathione sulfo-transferase (GST), Heme oxygenase (HO-1), Hydrogen peroxide (H_2_O_2_), Hydroxyl radical (OH-) Interleukin-1β (IL-1β), Kelch-like ECH-associated protein (Keap-1), Lipid peroxidase (LPO), Myeloperoxidase (MPO), Nitric oxide (NO), Nuclear factor kappa B (NF-κB), Nuclear factor erythroid 2-related factor 2 (Nrf-2), Superoxide radical (O2-.), Reactive oxygen species (ROS), Superoxide dismutase (SOD), Toll-like receptor (TLR-4), and Tumor necrosis factor-α (TNF-α).

**Table 1 ijms-22-06603-t001:** Peak intensities (a.u) of proteins, lipids, phospholipids and nucleic acids of wild-type, 5xFAD Tg and Bergenin-treated groups.

		Wild Type (a.u)	5xFAD Tg (a.u)	Bergenin (a.u)
**Proteins**	Amide-A	0.142 ± 0.005	0.047 ± 0.005 ^###^	0.107 ± 0.01 ***
	Amide-I	0.273 ± 0.01	0.083 ± 0.002 ^###^	0.213 ± 0.005 ***
	Amide-II	0.24 ± 0.002	0.073 ± 0.002 ^###^	0.174 ± 0.002 ***
	CH3 symmetric stretching	0.103 ± 0.02	0.03 ± 0.01 ^###^	0.081 ± 0.001 ***
**Lipids**	Olefinic=C-H stretching	0.075 ± 0.001	0.035 ± 0.003 ^###^	0.065 ± 0.05 ***
	CH3 asymmetric stretching	0.123 ± 0.005	0.046 ± 0.003 ^###^	0.084 ± 0.001 ***
	CH2 asymmetric stretching	0.16 ± 0.017	0.052 ± 0.01 ^###^	0.127 ± 0.02 ***
	CH2 symmetric stretching	0.156 ± 0.01	0.056 ± 0.01 ^###^	0.125 ± 0.021 ***
**Phospholipids**	PO^2−^ asymmetric stretch	0.21 ± 0.01	0.042 ± 0.002 ^###^	0.073 ± 0.02 ***
**Nucleic acids**	PO^2−^ symmetric stretch	0.226 ± 0.02	0.076 ± 0.002 ^###^	0.172 ± 0.002 ***

The data is presented as the mean (*n* = 6) ± SD. (###) denotes comparison to the wild-type control group. *** *p* < 0.001 denotes comparison to transgenic group.

**Table 2 ijms-22-06603-t002:** FT-IR absorption band area ratio for selected bands of wild-type, 5xFAD, Tg and Bergenin-treated hippocampal regions of mouse brains.

Lipid/Protein Ratio	Wild Type (a.u)	5XFAD Tg (a.u)	Bergenin (a.u)
Antiparallel β-sheet (I_1744_/I_1695_)	0.53 ± 0.02	0.77 ± 0.005 ^###^	0.67 ± 0.005 ***
Parallel β-sheet (I_1744_/I_1629_)	0.23 ± 0.05	0.44 ± 0.002 ^###^	0.36 ± 0.008 ***

The data is presented as the mean (*n* = 6) ± SD. (###) denotes comparison to the wild-type control group. *** *p <* 0.001 denotes comparison to transgenic group.

**Table 3 ijms-22-06603-t003:** Molecular docking analysis of Bergenin.

Proteins	PDB ID	Binding Energy (Kcal/mol)	Hydrogen Bonds	Hydrogen Bond Amino Acids	Hydrophobic Interactions
**Keap-1**	1U6D	−7.8	6	Val606, Gly367, Ala607, Val418, Val465, Val467	Val420
**Nrf-2**	2FLU	−8.3	3	Val465, Val608, Ala466	Val369, Ala607
**HO-1**	1irm	−6.6	3	Arg85, Ala87, His84	Arg86, Pro80
**TLR-4**	3vq2	−7.2	3	Gln505, Ser480, Asn456	---
**IκB-α**	6y1j	−7.2	4	Glu182, Ser64,Glu133, Gln67	Arg60
**NF-κB**	1vkx	−8.8	4	Gln247, Arg246, DA18, DA17	Lys541, Lys572
**Bcl-2**	2W3L	−7.7	2	Arg26, Arg68	---
**Bax**	5W62	−5.8	4	Ile175, Asn73, Ser72, Glu69	---
**Casp-3**	2XYG	−6.1	6	Met44, Lys82, Gly45, Glu84, Tyr41, Tyr83, Glu43	---
**Aβ**	4mvl	−7.3	2	Lys30, GLN26	His28

**Table 4 ijms-22-06603-t004:** FTIR spectra general peak assignments of brain tissue in 4000–500 cm^−1^ spectral range according to the literature [30,65,66].

Wavenumbers (cm^−1^)	Definition of the Assignment
3304	Amide A: mainly N–H stretching of proteins
3012	Olefinic=C-H stretching: lipids
2958	CH3 asymmetric stretching: mainly lipids
2922	CH2 asymmetric stretching: mainly lipids
2872	CH3 symmetric stretching: mainly proteins
2851	CH2 symmetric stretching: mainly lipids
1739	Ester C=O stretch: lipids
1653	Amide 1: C=O stretching of proteins
1548	Amide II: N-H bending and C-N stretching of proteins
1249	PO^2−^ asymmetric stretch: mainly phospholipids
1057	PO^2−^ symmetric stretch: mainly nucleic acids

## Data Availability

The data presented in this study are available from the corresponding author upon reasonable request.

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
