# Peer review of "Alleviation of Memory Deficit by Bergenin via the Regulation of Reelin and Nrf-2/NF-κB Pathway in Transgenic Mouse Model"

_ijms, 2021, doi:10.3390/ijms22126603_

Round 1
Reviewer 1 Report
Here the authors have applied a compound purified from the winter begonia / Fringed Bergenia named Bergenin to assays related to oxidative stress and survival in cell lines and behaviour and pathology in a mouse model of Alzheimer's disease.
This particular compound has already been suggested to alleviate/modulate many biological pathways and there are some credible studies in the literature that consider mechanistic pathways.
In this study, there are some effects shown after administration of the purified extract, on oxidative stress-related enzymes; however, I have some major concerns, plus the causation between the expression changes and behaviour and oral administration has not been demonstrated.
1) The purification quality or method is not shown, only referring to a patent.
2) The behavioural testing is potentially interesting, but very limited to two tests that could be confounded by many factors not considered, such as overall activity, motivation, anxiety, etc. further testing is needed to state a memory deficit has been alleviated.
3) The neuropathology is generally not high quality, with the sections shown not matching anatomically or any indication of where some of the regions or cited lesions are. Many of the sections are torn and damaged that cannot be suitable for quantification.
4) The docking data are purely speculative with no experimental follow-up
5) There is no data provided regarding how / where the compound is being received by the CNS or PNS / BBB / kinetics etc. given oral administration
Author Response
Point to point comments and response
Reviewer 1
Here the authors have applied a compound purified from the winter begonia / Fringed Bergenia named Bergenin to assays related to oxidative stress and survival in cell lines and behaviour and pathology in a mouse model of Alzheimer's disease.
This particular compound has already been suggested to alleviate/modulate many biological pathways and there are some credible studies in the literature that consider mechanistic pathways. In this study, there are some effects shown after administration of the purified extract, on oxidative stress-related enzymes; however, I have some major concerns, plus the causation between the expression changes and behaviour and oral administration has not been demonstrated.
Major comments
Comment 1
The purification quality or method is not shown, only referring to a patent.
Response
According to the reviewer’s recommendation, the method for purification of Bergenin was added and highlighted in section 3.2 of the revised manuscript.
Comment 2
The behavioural testing is potentially interesting, but very limited to two tests that could be confounded by many factors not considered, such as overall activity, motivation, anxiety, etc. further testing is needed to state a memory deficit has been alleviated.
Response
The reviewer is highly appreciated for the valuable comment. In order to determine the effect of Bergenin on spatial memory and locomotion, Morris water maze (MWM) and open field test (OFT) were performed. The MWM test has been widely used for the assessment of the spatial memory [1] and OFT was performed to determine the locomotor, anxiety and exploratory behavior in the 5xFAD Tg mice [2]. The results obtained from the activity were added and discussed accordingly in the revised manuscript. Results were shown in Figure 2 and 3 of the revised manuscript.
- Wenk, G. L., Assessment of spatial memory using the radial arm maze and Morris water maze. Current protocols in neuroscience 2004, 26, (1), 8.5 A. 1-8.5 A. 12.
- Sestakova, N.; Puzserova, A.; Kluknavsky, M.; Bernatova, I., Determination of motor activity and anxiety-related behaviour in rodents: methodological aspects and role of nitric oxide. Interdisciplinary toxicology 2013, 6, (3), 126-135.
Comment 3
The neuropathology is generally not high quality, with the sections shown not matching anatomically or any indication of where some of the regions or cited lesions are. Many of the sections are torn and damaged that cannot be suitable for quantification.
Response
The quality of the neuropathology was improved and all the cited regions were ensured to match the selected sections in Figure 4, 5 and 6 of the revised manuscript.
Comment 4
The docking data are purely speculative with no experimental follow-up
Response
The experimental data completely supported the docking data. The ligand (Bergenin)-protein interactions for all the targets mentioned in the present study were consistent with the results obtained by western blot and immunohistochemical (IHC) analysis.
Comment 5
There is no data provided regarding how / where the compound is being received by the CNS or PNS / BBB / kinetics etc. given oral administration
Response
The compound under study i.e., Bergenin has well established pharmacokinetic profile followed by oral administration. The study is supported by already reported, well-developed and validated liquid chromatography-mass spectrometry (LC-MS/MS) and high performance liquid chromatography (HPLC) method for the determination of Bergenin in animal plasma after oral administration [1-3]. Additionally, oral administration of Bergenin showed neuroprotective activity against neurological disorders in various animal models [4-7].
- Li, B.-H.; Wu, J.-D.; Li, X.-L., LC–MS/MS determination and pharmacokinetic study of bergenin, the main bioactive component of Bergenia purpurascens after oral administration in rats. Journal of pharmaceutical analysis 2013, 3, (4), 229-234.
- Yu, X.-a.; Teye Azietaku, J.; Li, J.; Wang, H.; Zheng, F.; Hao, J.; Chang, Y.-x., Simultaneous quantification of gallic acid, bergenin, epicatechin, epicatechin gallate, isoquercitrin, and quercetin-3-rhamnoside in rat plasma by LC-MS/MS method and its application to pharmacokinetics after oral administration of Ardisia japonica extract. Evidence-Based Complementary and Alternative Medicine 2018,
- Qin, X.; Zhou, D.; Huang, Y., [Kinetics study on intestinal absorption of bergenin in rats]. Sichuan da xue xue bao. Yi xue ban = Journal of Sichuan University. Medical science edition 2007, 38, (6), 1013-6.
- Barai, P.; Raval, N.; Acharya, S.; Borisa, A.; Bhatt, H.; Acharya, N., Neuroprotective effects of bergenin in Alzheimer’s disease: Investigation through molecular docking, in vitro and in vivo studies. Behavioural brain research 2019, 356, 18-40.
- Ji, Y.; Wang, D.; Zhang, B.; Lu, H., Bergenin ameliorates MPTP-induced Parkinson’s disease by activating PI3K/Akt signaling pathway. Journal of Alzheimer's Disease 2019, 72, (3), 823-833.
- Singh, J.; Kumar, A.; Sharma, A., Antianxiety activity guided isolation and characterization of bergenin from Caesalpinia digyna Rottler roots. J Ethnopharmacol 2017, 195, 182-187.
- Kumar, D.; Kumar, S., Neuroprotective constituents of Actaea acuminata (Wall. ex Royle) H. Hara roots. Zeitschrift fur Naturforschung. C, Journal of biosciences 2020.

Reviewer 2 Report
This is a very nice research article, which can be published as is.
Author Response
Thanks for positive comments on our paper
Reviewer 3 Report
The authors showed that the natural compound Bergenin displays neuroprotective properties in vitro and in vivo in Alzheimer’s disease (AD) mouse model in their comprehensive work. The authors show that Bergenin exerts its neuroprotective effect via regulating the Reeling signaling pathway, decreasing oxidative stress, neuroinflammation, and apoptosis. As a result, Bergenin alleviates spatial memory deficit in the AD model. Thus, Bergenin may represent a novel AD modifying therapeutics.
There are minor points.
- For consistency, in the introduction, the authors need to say a few words about how nitrosative stress is increased in AD.
- At the end of the Discussion section, I suggest adding a paragraph summarizing the key results obtained by the authors.
Author Response
Point to point comments and response
Reviewer 3
The authors showed that the natural compound Bergenin displays neuroprotective properties in vitro and in vivo in Alzheimer’s disease (AD) mouse model in their comprehensive work. The authors show that Bergenin exerts its neuroprotective effect via regulating the Reeling signaling pathway, decreasing oxidative stress, neuroinflammation, and apoptosis. As a result, Bergenin alleviates spatial memory deficit in the AD model. Thus, Bergenin may represent a novel AD modifying therapeutics. There are minor points.
Minor comments
Comment 1:
For consistency, in the introduction, the authors need to say a few words about how nitrosative stress is increased in AD.
Response
According to the reviewer’s recommendation, few sentences to explain the involvement of nitrosative stress in AD were added and highlighted in the introduction section of the revised manuscript.
Comment 2:
At the end of the Discussion section, I suggest adding a paragraph summarizing the key results obtained by the authors.
Response
According to the reviewer’s suggestion, a paragraph summarizing the key results obtained in the present study were added and highlighted in the discussion section of the revised manuscript.
